# Temporal processing and context dependency in *Caenorhabditis elegans* response to mechanosensation

**Mochi Liu[1], Anuj K Sharma[2], Joshua W Shaevitz[1,2], Andrew M Leifer[2,3]\***

[1]Lewis-Sigler Institute for Integrative Genomics, Princeton University, New Jersey, United States; [2]Department of Physics, Princeton University, New Jersey, United States; [3]Princeton Neuroscience Institute, Princeton University, New Jersey, United States

**Abstract** A quantitative understanding of how sensory signals are transformed into motor outputs places useful constraints on brain function and helps to reveal the brain's underlying computations. We investigate how the nematode *Caenorhabditis elegans* responds to time-varying mechanosensory signals using a high-throughput optogenetic assay and automated behavior quantification. We find that the behavioral response is tuned to temporal properties of mechanosensory signals, such as their integral and derivative, that extend over many seconds. Mechanosensory signals, even in the same neurons, can be tailored to elicit different behavioral responses. Moreover, we find that the animal's response also depends on its behavioral context. Most dramatically, the animal ignores all tested mechanosensory stimuli during turns. Finally, we present a linear-nonlinear model that predicts the animal's behavioral response to stimulus.
DOI: https://doi.org/10.7554/eLife.36419.001

## Introduction

An animal's nervous system interprets sensory signals to guide behavior, including behaviors that are involved in evading predation. Investigating how the nervous system processes these signals is a critical step towards understanding neural function.

Mechanosensation in the nematode *Caenorhabditis elegans* is an attractive platform for investigating sensorimotor processing. Six soft-touch mechanosensory neurons arranged throughout the body detect mechanical stimuli including those delivered either by a small probe in what is called a touch or by striking the petri dish containing the animal in what is called a tap (***Chalfie and Sulston, 1981***). Despite decades of investigation, however, the behavioral response to dynamic time-varying mechanosensory signals has not been fully explored.

Here we provide new details about the mechanosensory response system by quantitatively exploring the animal's detailed behavioral response to rich, dynamically varying signals. We find that the animal responds to the temporal features of signals in its mechanosensory neurons, such as its time-derivative (i.e. rate of change), that extend over many seconds. Moreover, we find evidence that the animal's sensorimotor response depends on the animal's current behavior state. That we find evidence of temporal processing and context dependency, even in the nematode's relatively simple touch circuit, raises the possibility that these features could be ubiquitous across sensory systems. Finally, we present a simple quantitative model that predicts the animal's response to novel mechanosensory signals.

Mechanosensation is important for *C. elegans* survival. *Caenorhabditis elegans* are preyed upon by nematophagous fungi, and touch-defective animals fail to detect and escape from the fungus (***Maguire et al., 2011***). Much is already known about this critical circuit. The six soft-touch

**\*For correspondence:**
leifer@princeton.edu

**Competing interests:** The authors declare that no competing interests exist.

**eLife digest** A worm called *Caenorhabditis elegans* has a nervous system made up of only 302 neurons, far fewer than the billions of cells that comprise our own brains. And yet these few hundred neurons are enough for these worms to detect and respond to their surroundings. *C. elegans* is thus a popular choice for studying how nervous systems process sensory information and use it to control behavior. Yet, most experiments to date have used only simple stimuli, such as taps or pokes, and studied a handful of behaviors, such as whether or not a worm stops moving or backs up. This limits the conclusions it has been possible to draw.

Liu et al. therefore set out to determine how the worm's nervous system responds to more complex stimuli. These included physical stimuli, such as taps on the side of the dish containing the worms, as well as simulated stimuli. To generate the latter, Liu et al. used a technique called optogenetics to directly activate the neurons in the worm's body that would normally detect information from the senses, by simply shining a light on the worms. Doing so gives the worm the sensation of a physical stimulus, even though none was present. Liu et al. then used mathematics to examine the relationships between the stimuli and the worms' responses.

The results confirmed that worms usually respond to simple stimuli, such as taps on the side of their dish, by backing up. But they also revealed more advanced forms of stimulus processing. The worms responded differently to stimuli that increased over time versus decreased, for example. A worm's response to a stimulus also varied depending on what the worm was doing at the time. Worms that were in the middle of turns, for instance, ignored stimuli to which they would normally respond. This suggests that an animal's current behavior influences how its nervous system interprets sensory information.

The discovery of relatively sophisticated responses to sensory stimuli in *C. elegans* indicates that even simple nervous systems are capable of flexible sensory processing. This lays a foundation for understanding how neural circuits interpret sensory signals. Building on this work will ultimately help us understand how more complicated nervous systems interpret and respond to the world.
DOI: https://doi.org/10.7554/eLife.36419.002

mechanosensory neurons detect both spatially localized and non-localized stimuli. Anterior touches are detected by anterior neurons ALML, ALMR and AVM and evoke reversal behaviors whereas posterior touches are detected by posterior neurons PLML and PLMR and evoke forward sprints (*Chalfie and Sulston, 1981*; *Chalfie et al., 1985*; *McClanahan et al., 2017*; *Mazzochette et al., 2018*). Non-spatially localized plate taps are detected by both anterior and posterior soft-touch neurons and evoke reversals in young adult animals (*Chalfie and Sulston, 1981*; *Rankin et al., 1990*); on rare occasions, they also evoke forward acceleration (*Wicks and Rankin, 1995*; *Chiba and Rankin, 1990*). Owing in part to its ease of delivery and its inherent compatibility with high-throughput methods (*Swierczek et al., 2011*), plate tap emerged early on as an assay for studying sensitization and habituation (*Rankin et al., 1990*). Plate tap has been used in concert with the touch assay to study the development, circuitry (*Chalfie and Sulston, 1981*), genes, molecules and receptors (*Sanyal et al., 2004*; *Kindt et al., 2007*) of the mechanosensory system.

When the animal interacts with its environment or brushes up against a nematophagous fungi's constricting ring, it necessarily receives time-varying stimuli. The response of an individual touch receptor neuron to force (*O'Hagan et al., 2005*), including to time-varying stimuli, is well characterized (*Eastwood et al., 2015*). The onset and offset of an applied force evokes strong excitatory currents that adapt with a timescale of a few tens of milliseconds (*O'Hagan et al., 2005*) and have a frequency response thought to peak in the 100 to 500 Hz range (*Eastwood et al., 2015*). Intracellular calcium activity in individual soft touch neurons has also been well characterized in response to touch and this activity exhibits slower transients that occur with a timescale of seconds (*Suzuki et al., 2003*; *Cho et al., 2018*). In contrast to this detailed understanding at the single neuron level, the animal's downstream response to rich temporally varying mechanosensory signals has been less well characterized.

The animal's behavior response to mechanosensory stimuli has primarily been studied using impulse stimuli. Specifically, the stimuli were either a brief application of touch, tap or optogenetic

stimulation, and the most salient feature of these stimuli was their amplitude, not their temporal profile (*Petzold et al., 2013*; *Stirman et al., 2011*; *McClanahan et al., 2017*; *Mazzochette et al., 2018*). In the classical touch assay, for example, a saturating force that lasts just a few tenths of a second is applied (*Nekimken et al., 2017*). Tap stimuli are even shorter in duration.

To our knowledge, the only temporally varying stimuli used to investigate behavioral responses to mechanosensation are: trains of taps or touches (*Chiba and Rankin, 1990*; *Kitamura et al., 2001*), trains of optogenetic pulses (*Porto et al., 2017*; *Leifer et al., 2011*), trains of ultrasound pulses (*Kubanek et al., 2018*), the delivery of 100 Hz or 1 kHz acoustic vibration (*Nagy et al., 2014a, 2014b*; *Sugi et al., 2016*), and the delivery of sustained acoustic vibrations of different frequencies lasting many minutes to hours (*Chen and Chalfie, 2014*).

The following behaviors have been extensively studied in response to mechanosensory stimulation. Early work scored the animal's reversals (*Chiba and Rankin, 1990*) and more recent work includes reversal distance (*Kitamura et al., 2001*), rate of reversals (*Swierczek et al., 2011*) or pauses, reversal duration and reversal latency (*Ardiel et al., 2017*). The effect of mechanosensory stimulation on accelerations has also been studied (*Wicks and Rankin, 1995*). Recent work, however, shows that the animal's repertoire of behavior is larger (*Stephens et al., 2008*; *Brown et al., 2013*).

Over short timescales, reversals or accelerations depend on the set of neurons stimulated and the stimulus strength. The location of an applied force determines which touch receptor neurons are activated and thus whether the animal accelerates or reverses, while the amplitude of the applied stimulus determines the probability that the animal responds at all (*Driscoll, 1997*; *Stirman et al., 2011*; *Petzold et al., 2013*; *McClanahan et al., 2017*; *Mazzochette et al., 2018*).

Over longer timescales of minutes to hours, however, the picture has been shown to be more complicated. Habituation (*Rankin et al., 1990*), quiescence (*Raizen et al., 2008*; *Cho et al., 2018*), and exposure to prolonged vibrations, salt or hypoxia, all modulate the animal's sensitivity to mechanical stimuli (*Chen and Chalfie, 2014, 2015*).

More recently, evidence has also emerged that short timescale properties of the stimulus may also play a role in modulating the animal's behavioral response. *Porto et al. (2017)* reported the use of reverse correlation and a binary optogenetic stimulus to present evidence that temporal processing is important for the animal's behavioral response over a timescale of seconds. In our work here, we show that the nervous system does indeed process signals from the mechanosensory neurons over timeseries of many seconds. We find that the animal's behavior response depends on higher-order temporal features such as the derivative of those mechanosensory signals, in addition to the stimulus amplitude and the animal's own behavioral context.

Here, we revisit the animal's behavioral response to mechanosensory stimulation armed with high-throughput optogenetic methods for delivering time-varying stimuli and improved techniques for measuring animal posture (*Stephens et al., 2008*) and behavior (*Berman et al., 2014*). Using reverse correlation (*Ringach and Shapley, 2004*; *Schwartz et al., 2006*; *Gepner et al., 2015*), we analyze over 8000 animal-hours of recordings and find new insights into the interplay between sensory processing and behavior.

## Results

### Mechanosensation evokes a range of behavioral responses

We first investigated the animal's response to plate tap, a spatially non-localized mechanosensory stimulus generated by tapping the dish containing the animals. Plate taps had previously been reported to evoke reverse locomotion (*Rankin et al., 1990*) and rarely forward accelerations (*Wicks and Rankin, 1995*) in the young adult animals used here. A solenoid repeatedly delivered a tap stimulus every 60 s for 30 min to a plate of many young adult wild-type (N2) worms, repeated across 22 plates, resulting in 40,409 total animal-tap presentations. The inter-stimulus interval was chosen to minimize the effects of habituation (*Rankin and Wicks, 2000*). The animal's behavior was continuously measured and classified using a behavior-mapping technique similar to that described in *Berman et al. (2014)*. Briefly, statistical inference was performed on all of the animal's posture dynamics to generate a single behavior map. Stereotyped posture dynamics that emerged from this map were defined as behaviors. Each individual animal's posture dynamics were projected into this map at each point in time and automatically classified into one of nine behavior states, which were

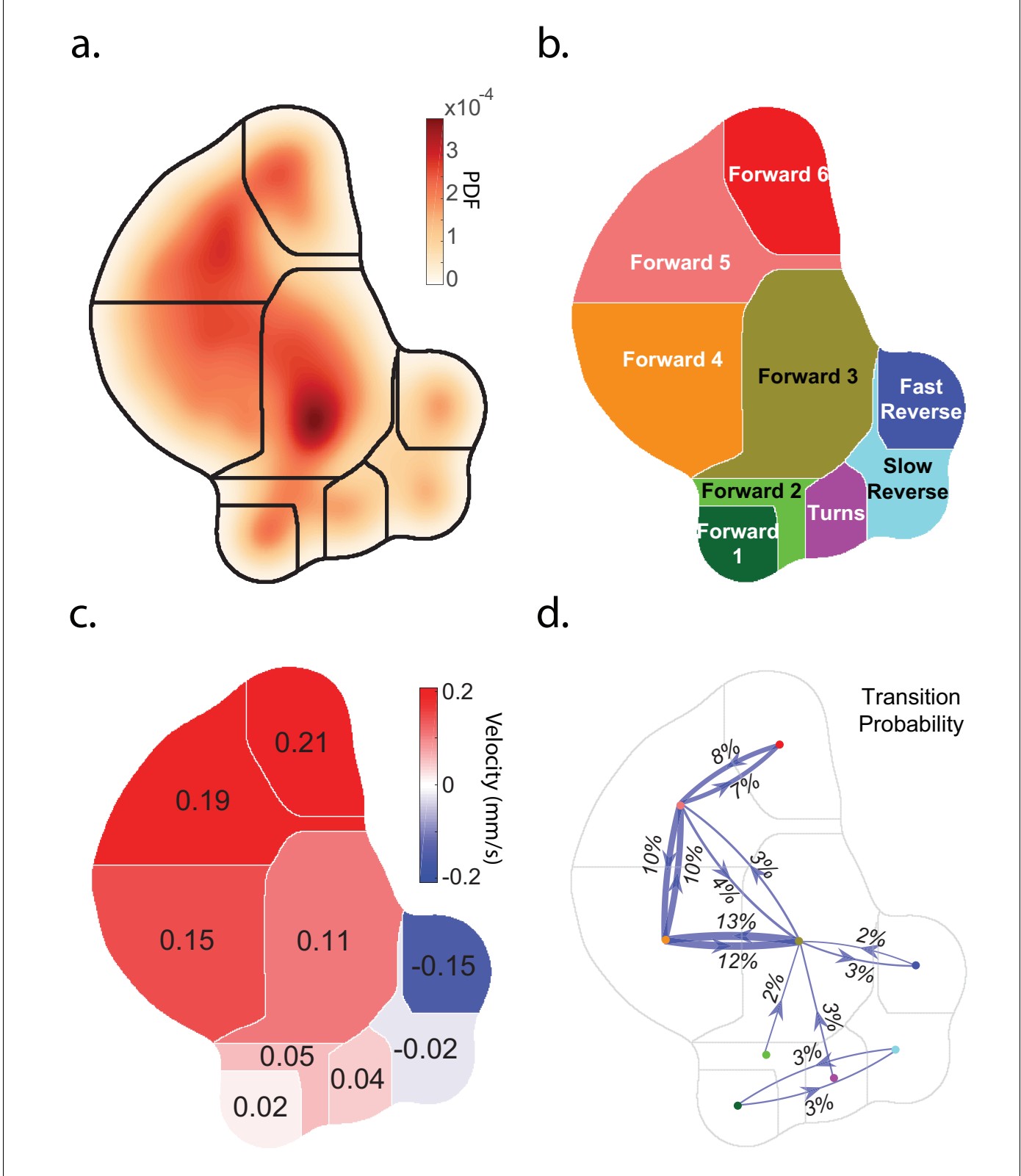

**Figure 1.** *Caenorhabditis elegans* behavior quantification. (a) Behavior map showing the probability density of posture dynamics observed during 2284 animal-hours of behavior, including stimulus and control conditions ('Random Noise' row in *Table 2*). Posture dynamics have many dimensions but are projected down into a low-dimensional space using the t-SNE method used by *Berman et al. (2014)*. Peaks indicate stereotyped postures. Discrete behavior states are defined by dividing the posture map into nine regions by using a watershedding algorithm. (b) Human-readable behavior names

*Figure 1 continued on next page*

*Figure 1 continued*

are provided by the experimenters. (c) Mean center of mass velocities of animals in each region. Positive velocity is in the direction of the animal's head. (d) Probability of transitioning between behaviors. Thickness of lines scales with probability. Transition probabilities < 2% were omitted.

DOI: https://doi.org/10.7554/eLife.36419.003

The following video and figure supplements are available for figure 1:

**Figure supplement 1.** Analysis pipeline for classifying behavior.

DOI: https://doi.org/10.7554/eLife.36419.004

**Figure supplement 2.** Behavior maps were generated from 2284 animal-hours of behavior recorded from P*mec-4::Chrimson* worms during optogenetic stimulation and control conditions.

DOI: https://doi.org/10.7554/eLife.36419.005

**Figure 1—video 1.** Video of tracked animals undergoing optogenetic stimulation.

DOI: https://doi.org/10.7554/eLife.36419.006

**Figure 1—video 2.** Videos of randomly selected animals performing each of the nine behaviors.

DOI: https://doi.org/10.7554/eLife.36419.007

**Figure 1—video 3.** Video showing the path of an animal through behavior space.

DOI: https://doi.org/10.7554/eLife.36419.008

assigned labels such as 'Turn.' See *Figure 1*, *Figure 1—figure supplements 1* and *2* and methods for a complete description of the behavior mapping. Also see example videos of behavioral mapping in *Figure 1—video 2* and *Figure 1—video 3*.

Consistent with previous reports, we observed that taps most dramatically evoked the animal to transition to the 'Fast Reverse' state. Tap stimulus induced a 14-fold increase in the fraction of animals exhibiting 'Fast Reverse' immediately post stimuli, see *Figure 2a* and *Figure 2—figure supplement 2*. In addition, animals that continued in forward locomotion exhibited an overall slowing down, transitioning from fast locomotion states to slower locomotion states, which to our knowledge had not been reported previously. We also observed a 4.5-fold increase in the fraction of animals exhibiting 'Turn' behavior approximately 5 s post stimulus. The fraction of animals exhibiting 'Slow Reverse' also increased slightly upon stimulation. These measurements suggest that plate tap evokes a wide-range of behavioral responses in the animal.

## Optogenetic stimulation mimics a tap

We sought to activate the mechanosensory circuit optogenetically because optogenetic stimulation is more amenable to modulation and control. Optogenetic stimulation of the six mechanosensory neurons had previously been shown to evoke reversals and accelerations, similar to the response to tap (*Nagel et al., 2005*). We wondered whether the details of the behavior response to tap that we observed are also present in response to optogenetic activation. Animals expressing the light-gated ion channel Chrimson in their soft touch mechanosensory neurons (strain AML67 [P*mec-4::Chrimson::SL2::mCherry::unc-54*]) were illuminated with red light for 1 s with a 60 s inter-stimulus interval (2,444 stimulus-animal presentations, 20 $\mu W\ mm^{-2}$, selected to be in a region of high behavior sensitivity, see *Figure 2c*). Consistent with previous reports, light stimulation evoked a behavior response that was quantitatively similar to that of the plate tap (see *Figure 2b*) and required the cofactor all-trans retinal (ATR), see *Figure 2—figure supplement 2*. For both light and tap, the most salient response was a dramatic increase in animals in the 'Fast Reverse' state. Both light and tap also evoked an increase in 'Forward 3' behaviors and both evoked similar decreases in 'Forward 4', '5' and '6' behaviors. Both light and tap also evoked an increase in 'Turn' behavior that peaked 5 s post-stimulus. Hence, optogenetic stimulation of mechanosensory neurons evoke detailed behavior responses similar to those resulting from a mechanical stimulus. This suggests that our optogenetic stimulation generates physiologically reasonable signals in the mechanosensory neurons and we therefore proceeded to explore the animal's response to optogenetic stimulation.

## Behavioral responses are correlated to temporal features such as the derivative

When the animal explores its natural environment, crawls through crevices, and interacts with other organisms, it probably experiences time-varying mechanical stimuli. Therefore, we sought to investigate the animal's response to random temporally varying optogenetic stimulation. We find that the

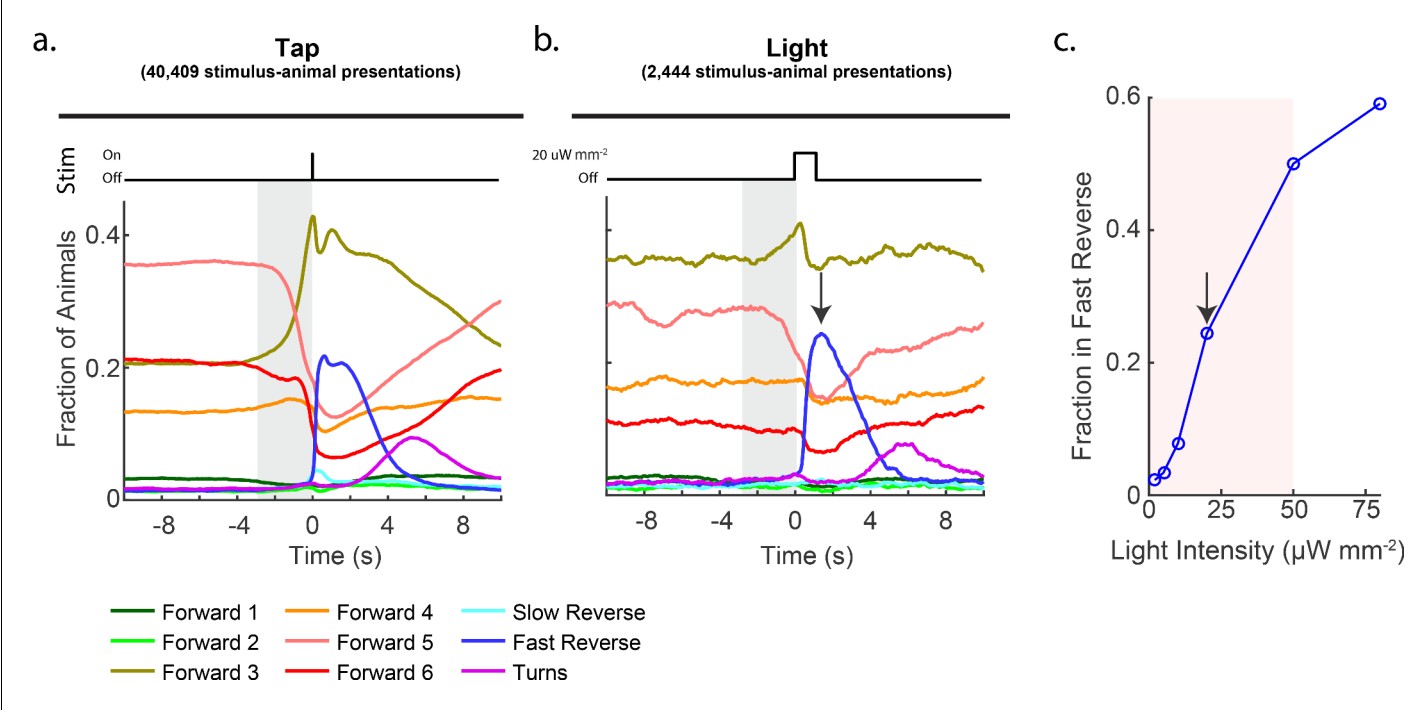

**Figure 2.** Stimulation evokes a diverse range of behavior responses. (a) Fractions of animals occupying each behavior state in response to a plate tap (40,409 stimulus-animal presentations) and (b) in response to a 1 s optogenetic light stimulation of the six soft touch mechanosensory neurons (2,444 stimulus-animal presentations, 20 µW mm$^{-2}$). Note the similarity in the behavior responses to light and tap. The gray shaded window indicates inherent temporal uncertainty in behavior classification. See 'Materials and methods'. (c) Response to optogenetic stimulation depends on light intensity. Peak fraction of animals in the 'Fast Reverse' state in a 6 s window post stimulus are shown for different-intensity light pulses. More than 2,000 stimulus-animal presentations were recorded for each point plotted. Arrow indicates the light intensity used in (B). Pink shaded region indicates light range used for subsequent continuous light stimulation experiments, as in *Figure 3*.

DOI: https://doi.org/10.7554/eLife.36419.009

The following figure supplements are available for figure 2:

**Figure supplement 1.** Diagram of high-throughput stimulation and behavior assay.

DOI: https://doi.org/10.7554/eLife.36419.010

**Figure supplement 2.** Transition rates for tap and light stimulation.

DOI: https://doi.org/10.7554/eLife.36419.011

**Figure supplement 3.** Control animals grown without ATR are light insensitive.

DOI: https://doi.org/10.7554/eLife.36419.012

**Figure supplement 4.** Tap sensitivity of transgenic animals is reduced compared to that of wild-type animals.

DOI: https://doi.org/10.7554/eLife.36419.013

animal's specific behavioral response correlates with higher-order temporal features of the stimulus, not merely the amplitude.

To deliver rich temporally varying stimuli, we continuously presented a plate of transgenic animals with light modulated by broad frequency noise (7 Hz nyquist limit, 0.5 s correlation time, 25 µW mm$^{-2}$ average intensity, min 0, max 50 µW mm$^{-2}$, see power spectra in *Figure 3—figure supplement 4*, and video in *Figure 1—video 1*).

Modulating light intensity has been shown to elicit graded potentials during optogenetic activation of other *C. elegans* neurons (*Liu et al., 2009*; *Narayan et al., 2011*), therefore we expect the time-varying light stimuli to result in a membrane potential that varies smoothly over time. Noise stimulation evoked a wide range of behavioral responses (see *Figure 3—figure supplement 1*). We used reverse correlation to identify the salient features of the stimulus that correlates with transitions into each behavior. Reverse correlation yields kernels that describe how a behavior is tuned to a stimulus. Kernels are particularly powerful in the context of the linear non-linear (LN) model, a simple and ubiquitous model in neuroscience that can be used to predict a neuron's or animal's response

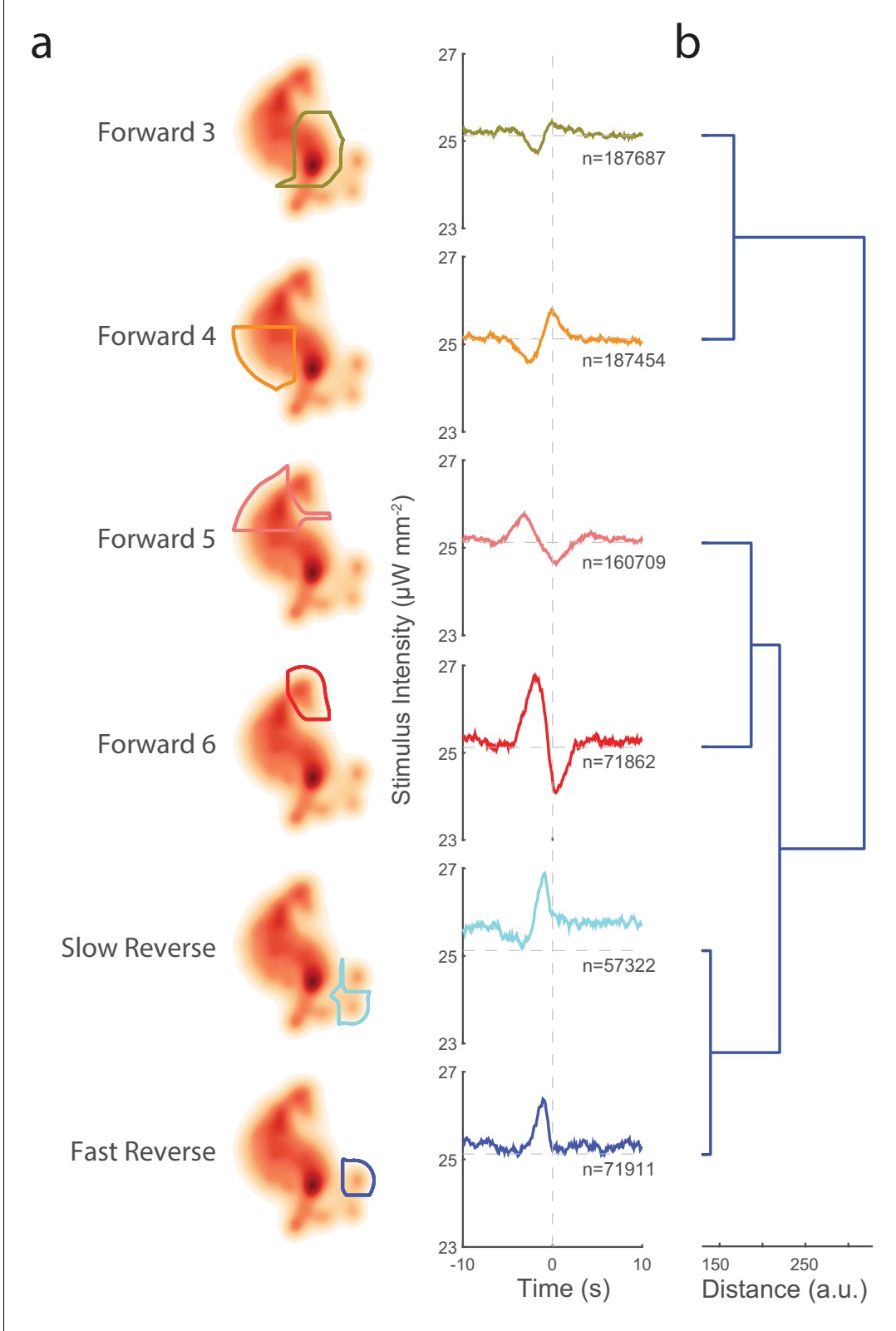

**Figure 3.** Transitions into behavior states are tuned to higher-order temporal features of the stimulus such as the derivative. (a) Random noise time-varying light stimulus is delivered to a population of animals. Behavior-triggered averages (also referred to as kernels) are calculated for transitions into each behavior state from 1,784 animal-hours of recordings. Each behavior-triggered average describes features of the stimulus that correlate with that behavior transition. Only those behavior-triggered averages that pass a significance test in which they are compared to a shuffled stimuli are shown.
*Figure 3 continued on next page*

*Figure 3 continued*

The shape of the behavior-triggered average depends on the behavior. Note that some behaviors have Gaussian-like shapes, whereas others have biphasic shapes that act like derivatives. The numbers of observed transitions, *n*, in each behavior are listed. (**b**) Similar behaviors have similar behavior-triggered averages. Dendrogram showing hierarchical clustering of the euclidian distance of the scaled behavior-triggered averages. The two reversal states, for example, form a cluster.

DOI: https://doi.org/10.7554/eLife.36419.014

The following figure supplements are available for figure 3:

**Figure supplement 1.** Change in behavioral occupancy evoked by random noise stimulation.

DOI: https://doi.org/10.7554/eLife.36419.015

**Figure supplement 2.** Behavior-triggered averages and non-linearities for all behaviors.

DOI: https://doi.org/10.7554/eLife.36419.016

**Figure supplement 3.** Behavior-triggered averages for control animals grown without ATR.

DOI: https://doi.org/10.7554/eLife.36419.017

**Figure supplement 4.** Power spectra of a single instantiation of the random noise stimulus.

DOI: https://doi.org/10.7554/eLife.36419.018

**Figure supplement 5.** Light-intensity histogram for random noise stimulus.

DOI: https://doi.org/10.7554/eLife.36419.019

to stimulus (*Ringach and Shapley, 2004*; *Schwartz et al., 2006*; *Coen et al., 2014*; *Gepner et al., 2015*; *Hernandez-Nunez et al., 2015*; *Calhoun and Murthy, 2017*; *Clemens and Murthy, 2017*). See in particular (*Gepner et al., 2015*). Briefly, the LN model treats the response to a stimulus as a stochastic process involving two steps: first the stimulus timeseries $s(t)$ is convolved with a kernel $A$ (linear operation), and then it is transformed into a response probability $P$ via a non-linear look-up function $f$ (non-linear operation), such that,

$$P[\text{behavior}](t) = f[(A^\star s)(t)]; (A^\star s) = \int_0^\infty A(\tau)s(t-\tau)d\tau. \tag{1}$$

The shapes of the kernel and the non-linearity describe how a behavior response is tuned to the stimulus.

Kernels can be estimated by finding the behavior-triggered average. Briefly, the stimulus in a time window centered on a behavior transition is averaged across all such behavior transitions. The mean subtracted and time-reversed behavior-triggered average is an estimate of the kernel, and so henceforth, we use the terms behavior-triggered average and kernel interchangeably. Once the kernels $A$ are calculated, it is straightforward to estimate the non-linearities $f$ from the observed behavior responses (see 'Materials and methods'). Kernels and associated non-linearities were computed for transitions into each of the nine behavior states from over 50,000 behavior transition events per behavior (see *Figure 3* and *Figure 3—figure supplement 2*). Kernels for six of the nine behaviors were found to be significant compared to a shuffled stimuli (see 'Materials and methods'). By contrast, kernels computed from control animals grown without the necessary cofactor ATR all failed to pass our significance threshold (see *Figure 3—figure supplement 3*). Non-linearities calculated for the six behaviors were found to be mostly linear, suggesting that in our case the kernels themselves capture most of the information about how the nervous system responds to our stimulus.

Our prior understanding of the mechanosensory circuit makes strong predictions about the shape of the kernels that we should expect. If the behavior depends only on which neurons are activated, then all kernels should have the same shape, scaled linearly, because we are always activating the same set of neurons. (This assumes that all six neurons are activated in a linear regime, which seems reasonable given the approximately linear response observed in *Figure 2c*). Moreover, if the probability of response depends only on instantaneous stimulus amplitude, then we further expect all kernels to be narrow Gaussians. In contrast to these predictions, we see a wide diversity of kernels. Forward-locomotion kernels have biphasic waveforms, not at all like Gaussians. 'Forward 6', for example, has the shape of a differentiator, suggesting that the transitions into 'Forward 6' correlate with decreasing stimuli on a 7 s timescale. Kernels for 'Slow Reverse' and 'Fast Reverse', on the other hand, do look like Gaussians, consistent with prior reports that *reversals* do depend on the stimulus amplitude. Interestingly, the Gaussians are wide, which suggests that the animal may integrate the sensory signal over approximately 3 s in determining to reverse.

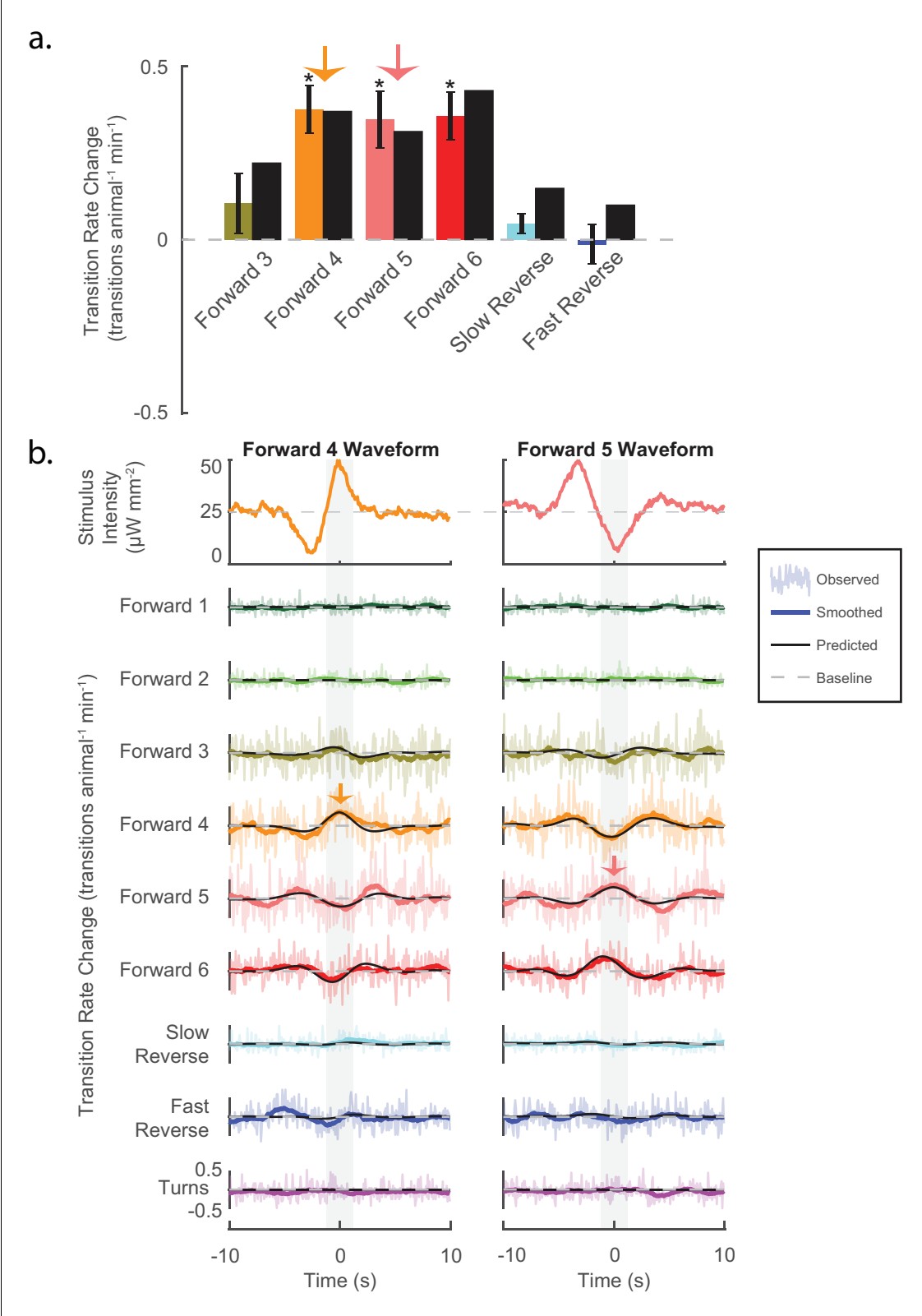

**Figure 4.** Stimuli can be tailored to elicit specific behavioral responses, and the LN model predicts such responses. (a) Animals are presented with stimuli shaped like the kernels in *Figure 3*. Predicted (black bar) and observed (color bar) changes in transition rate are shown for transitions into each kernel-shaped stimulus' corresponding behavior. For example, a 'Forward 3'-shaped stimulus increases transitions into 'Forward 3' (mustard bar). For five of the six behaviors, stimulation evoked increased transitions into their corresponding behaviors, as predicted. Transition rate changes are

*Figure 4 continued on next page*

*Figure 4 continued*

measured with respect to baseline (see 'Materials and methods'). Significance was estimated via a t-test and error bars show the standard error of the mean. The number of stimulus-animal presentations, from left to right, were 14,238, 13,612, 14,699, 14,424, 14,194 and 13,708. Of these, the number of timely transitions observed were 14,00, 1,428, 1,692, 944, 191 and 513. The p-values were 2.2e–1, 5.6e–6, 1e–4, 3.4e–5, 7.5e–2, 9.5e–1. (b) The LN model predicts details of the animal's behavioral response. For each point in time, the LN model predicts the change from baseline of transition rates for all nine behaviors in response to a stimulus. Detailed responses to 'Forward 4'- and 'Forward 5'-kernel-shaped stimuli are shown (see *Figure 4—figure supplement 1* for the rest). Raw transitions rates (light colored shading), smoothed transition rates (colored line) and LN prediction (solid black line) are shown. For stimuli that are shaped like 'Forward 4', the LN model correctly predicts not only that transitions into 'Forward 4' increase but also that transitions into 'Forward 5' and '6' decrease. Light gray shading indicates the 2 s time window used to calculate transition rates for the transitions shown in (A) (orange and pink arrows). Of 13,612 and 14,699 presentations for 'Forward 4'- and '5'-kernel shaped stimuli, respectively, the following number of transitions were observed in the 20 s window: by row for 'Forward 4'-shaped 1,265, 1,330, 12,312, 11,962, 13,436, 6,861, 1,735, 4,934 and 2,864, and for 'Forward 5'-shaped 1,198, 1,437, 13,657, 13,538, 14,656, 7,295, 1,673, 5,506 and 3,118.
DOI: https://doi.org/10.7554/eLife.36419.020

The following figure supplements are available for figure 4:

**Figure supplement 1.** Behavioral responses to all kernel-shaped stimuli.
DOI: https://doi.org/10.7554/eLife.36419.021

**Figure supplement 2.** Control animals grown without ATR do not respond to kernel-shaped stimuli.
DOI: https://doi.org/10.7554/eLife.36419.022

Taken together, we conclude that the animal's behavior response is not merely correlated with which neurons are stimulated and the stimulus amplitude. Instead different behaviors correlate with different temporal features of signals in the mechanosensory neurons, even though the same six neurons were always activated. The behavioral response correlates with properties of the stimulus such as the derivative or the integral, not just the amplitude.

## Similar behavioral responses are tuned to similar stimuli

We wondered about the organization of the behavioral responses with respect to the stimuli to which they are tuned. One might expect animals to have evolved their behavioral response so that similar behaviors are tuned to similar stimuli. Indeed, we find that similar behaviors have quantitatively similar kernels. Hierarchical clustering was performed on the euclidian distance of the scaled kernels (see *Figure 3*). The two reverse locomotion states have similar kernels and were clustered together. Forward velocity states fell into two clusters that were based on speed: 'Forward 3' and 'Forward 4' are slower and clustered together, whereas 'Forward 5' and 'Forward 6' are faster and clustered together. That similarities in the kernels reflect the similarities in their associated behaviors, provides additional confidence in our reverse correlation analysis.

## Stimuli can be tailored to generate specific behavioral responses

To test causally whether specific signals in the mechanosensory neurons can bias the animal towards specific behaviors as predicted, we generated stimuli that were tailored to elicit specific behavioral responses. The kernels found in *Figure 3* purport to describe how each behavioral response is tuned to the stimuli. Therefore, stimuli shaped like one of the kernels should drive an *increase* in transitions into its respective behavior. If, however, the behavioral response is tuned differently, then the kernel-shaped stimulus may evoke *decreases* in transitions to that behavior. (We already know that the animal can respond to some stimuli by decreasing transitions to certain behaviors because we saw this with tap and 'Forward 6', for example [see *Figure 2—figure supplement 2*]).

We tested whether stimuli that are shaped like the kernels in *Figure 3* increased transitions into their associated behaviors. Kernel waveforms were presented to a plate of animals in a randomized order (six kernels, >13,500 animal-stimulus presentations per kernel; 40 s inter-stimulus interval). Five of six kernels elicited increased transitions to their respective behaviors as predicted, three of the six significantly so (see *Figure 4a*). None significantly decreased transitions to their respective behaviors. We therefore conclude that the kernels correctly depict tuning of the behavioral responses. Consequently, we conclude that mechanosensory signals (even in the same neurons) can be tailored to evoke specific behaviors just by altering the stimulus waveform.

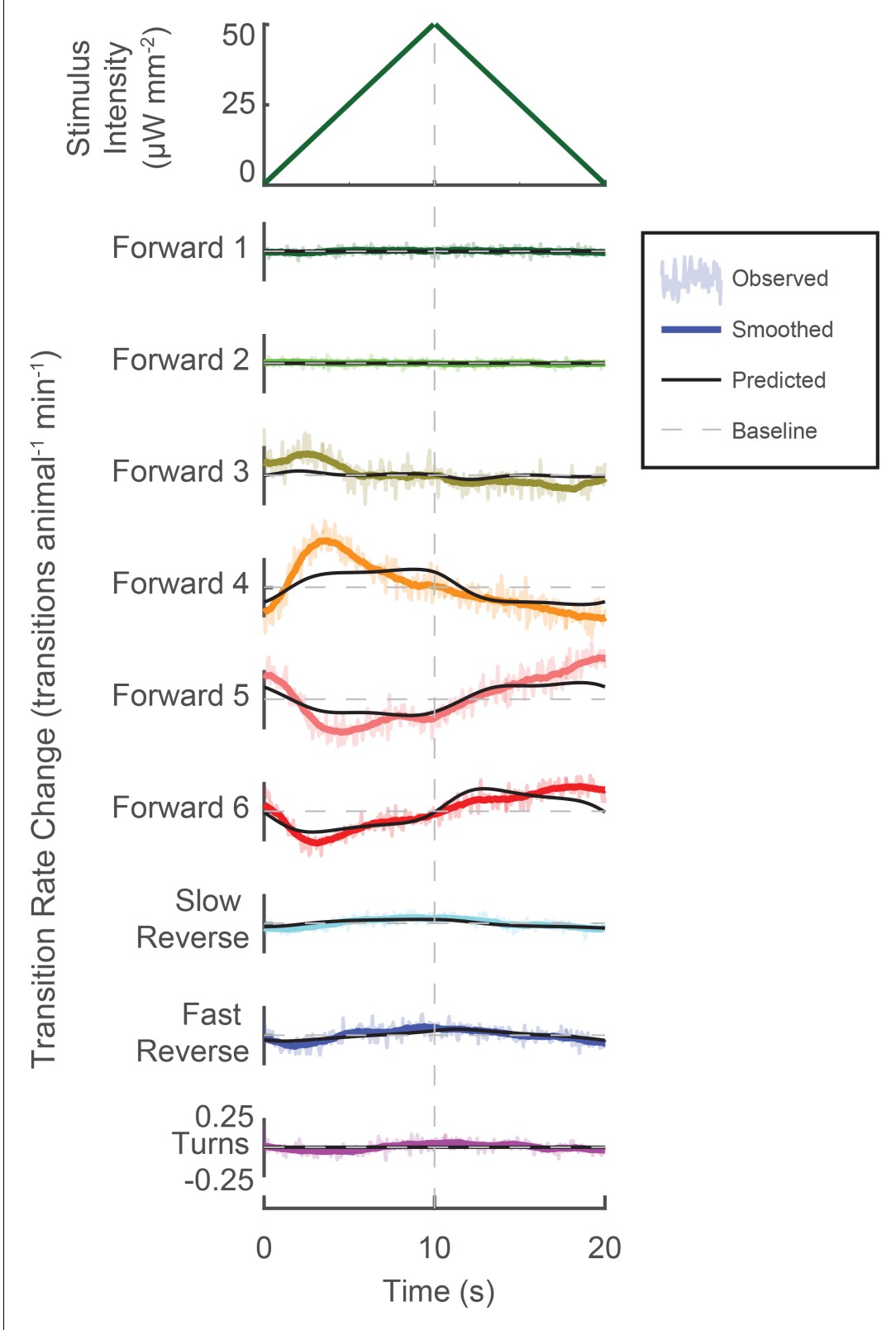

**Figure 5.** Novel stimuli can be constructed to enrich specific mechanosensory responses. A novel triangle-wave optogenetic light stimulus was repeatedly presented to animals. Change in transition rates are shown for transitions into each behavior (raw, light color shaded; smoothed, solid color line). Changes to transition rate as predicted by the LN model are also shown (black line). Increasing light intensity increases transitions into 'Forward 3'and 'Forward 4', while decreasing light intensity increases transitions into 'Forward 5' and 'Forward 6'. Transitions into 'Slow Reverse' and 'Fast

*Figure 5 continued on next page*

*Figure 5 continued*
Reverse' are highest during greatest stimulus intensity. The LN model predicts these trends (though not all the details) even though the LN model was fitted using the random noise experiments and therefore was not exposed to this particular stimulus. In response to the 340,757 animal-stimulus presentations, the following number of transitions were observed (by row, from top to bottom): 33,315, 31,243, 298,400, 343,474, 327,509, 160,332, 43,909, 106,743, and 57,439.
DOI: https://doi.org/10.7554/eLife.36419.023
The following figure supplement is available for figure 5:

**Figure supplement 1.** Control animals grown without ATR do not respond to triangle wave.
DOI: https://doi.org/10.7554/eLife.36419.024

## LN model predicts behavioral response, including response to novel stimuli

The LN model provides an analytical framework to predict how an animal responds to a stimulus. The LN model correctly predicted that kernel-shaped waveforms should increase transitions into each kernel's associated behavior state (see *Figure 4a*). The kernel-shaped waveforms also evoked other behavioral responses. For example, stimuli that were shaped like the 'Forward 4' kernel increased transitions to both 'Forward 4' and 'Forward 3'; but decreased transitions to 'Forward 5' and 'Forward 6' (see *Figure 4b*). How well, we wondered, does the LN model predict those responses? We compared the observed behavioral responses (colored lines) to detailed time-dependent predictions made by the LN model (black lines). To the resolution at which we could observe, we were reassured to find that the LN model correctly predicted the sign and temporal profile of changes in transition rates for all nine behavior states in response to each of the six kernel stimuli (see *Figure 4b* and *Figure 4—figure supplement 1*), suggesting that the LN model captures myriad details of the animal's behavioral response.

We further challenged our understanding of the animal's behavioral response to stimulus by presenting an entirely novel stimulus, a triangle-wave (340,757 stimulus-animal presentations) (see *Figure 5* and *Figure 5—figure supplement 1*). How well does the LN model predict the animal's behavior response to this novel stimulus? The LN model captured the sign and general trend (though not all features) of the time-dependent change in the transition rate to all nine behaviors in response to the triangle wave. Moreover, the LN model provides a framework for understanding the animal's response by inspecting features of the kernel waveform. For example, the 'Fast Reverse' kernel is symmetric in time and its mean-subtracted integral is positive. Therefore the shape of the 'Fast Reverse' kernel suggests that 'Fast Reverse' should be tuned to the overall stimulus intensity but not its derivative. Indeed we observe a very slight increase in the rate of transitions to 'Fast Reverse' during peak stimulus intensity. Conversely, the 'Forward 6' kernel is asymmetric in time and its biphasic waveform resembles that of the negative derivative of a Gaussian. Therefore, 'Forward 6' should be tuned to decreases in stimulus intensity, as we observe.

Taken together, our experiments show that the animal can be driven to transition into different specific behavior states by modulating the temporal profile of signals in the same mechanosensory neurons, and that the LN model predicts the animal's response.

## Sensory processing is context dependent

*Caenorhabditis elegans* are known to respond differently to the same stimuli when they are in different long-lived behavior states such as hunger (*Ghosh et al., 2016*), quiescence (*Raizen et al., 2008*; *Schwarz et al., 2011*; *Nagy et al., 2014b*; *Cho et al., 2018*) or arousal (*Cho and Sternberg, 2014*), or while undergoing Dauer formation (*Chen and Chalfie, 2014, 2015*). We wondered whether mechanosensory processing might also be influenced by short-lived behavior states, like the 'Turn', 'Reverse' or 'Forward' locomotory states measured here. To investigate tuning of the animal's behavioral response conditional on its current behavior state, we calculated context-dependent kernels, one for each pairwise transition (see *Figure 6—figure supplement 1*). Of 72 possible pairwise transitions, 27 had kernels that passed our shuffled significance threshold (compared to only four for our off-retinal control (see *Figure 6—figure supplement 2*). Transitions to some behavior states, such as 'Forward 4', had kernels that changed dramatically depending on which behavior the animal originated from (see columns in *Figure 6—figure supplement 1*). The pairwise-specific kernels

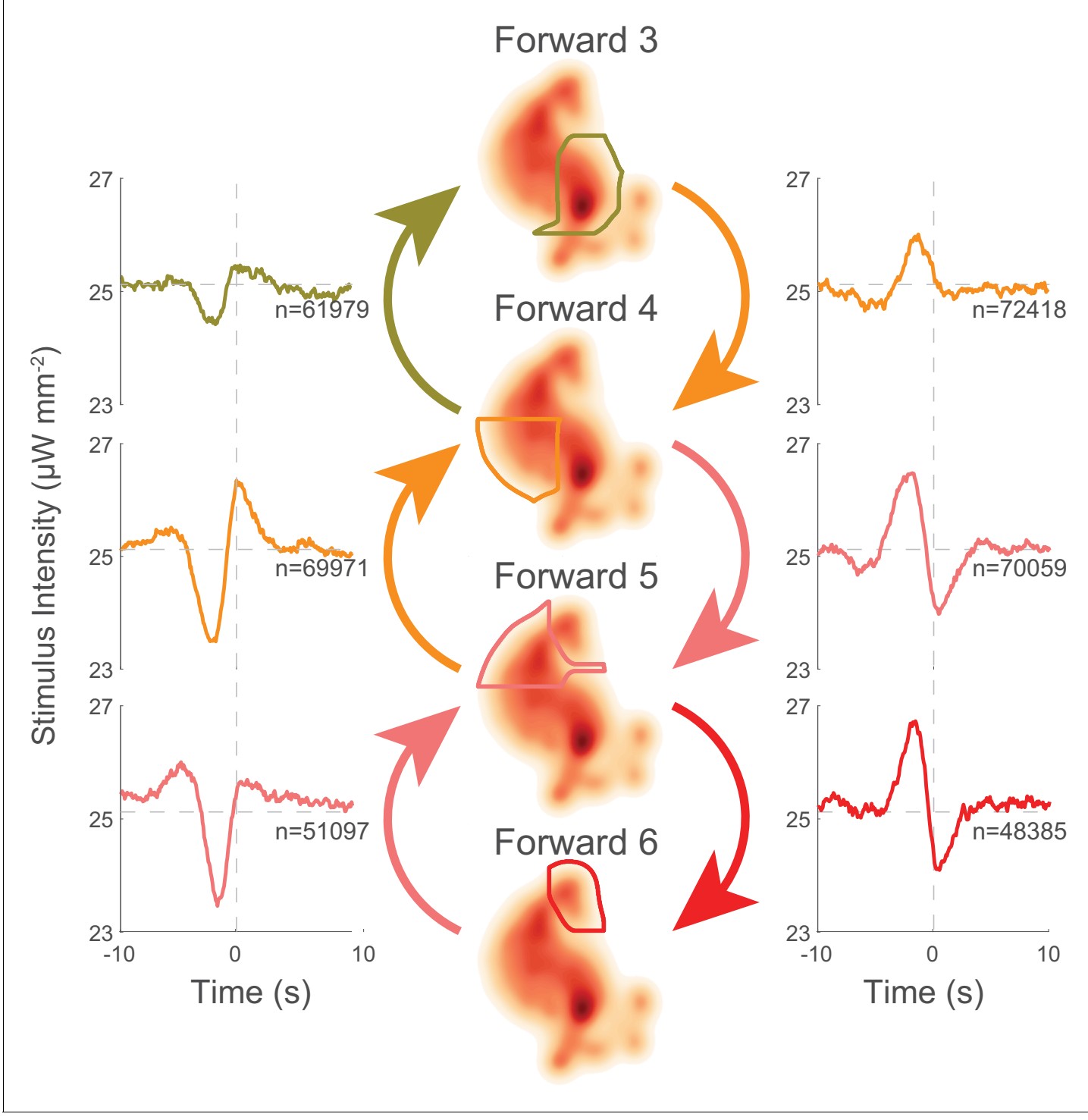

**Figure 6.** Behavior transitions that involve slowing down and speeding up have stereotyped tuning. Selected context-dependent kernels are shown for transitions amongst forward locomotory states, where higher numbered states have higher velocities. Kernels for slowing transitions (left column) are all similar, whereas kernels for speeding up transitions (right column) are also similar. Slowing and speeding-up kernels resemble horizontal reflections of one another.

DOI: https://doi.org/10.7554/eLife.36419.025

The following figure supplements are available for figure 6:

**Figure supplement 1.** All 72 pairwise context-dependent behavior-triggered averages.

DOI: https://doi.org/10.7554/eLife.36419.026

*Figure 6 continued*

**Figure supplement 2.** All 72 pairwise context-dependent behavior-triggered averages for control animals grown without ATR.
DOI: https://doi.org/10.7554/eLife.36419.027

provided evidence of two types of context-dependent sensory processing in *C. elegans* that occur within short-time scales. In both cases, the animal appears to respond to the same stimuli differently depending on its current behavior. In the first, the animal responds to certain mechanosensory signals by speeding up or slowing down. In the second type, the animal suppresses its response to mechanosensory stimuli during turning behavior. These two types of context-dependency are described below.

## There are mechanosensory signals for speeding up or slowing down

Behavior transitions that involve slowing down have similar tuning. For example, the 'Forward 5'→'Forward 4' kernel has a similar shape to the 'Forward 4'→'Forward 3' kernel (see *Figure 6*, left column). Likewise, transitions involving speeding up also have similar kernels. For example, 'Forward 3'→'Forward 4' and 'Forward 4'→'Forward 5' have similar kernels (see *Figure 6*, right column). Moreover, the two classes of kernels appear to be reflections of one another about the line of mean stimulus intensity. The stereotypy of the speed up and slow down kernels suggests that these nematodes have evolved to respond to certain stimuli by slowing down or speeding up in a relative way

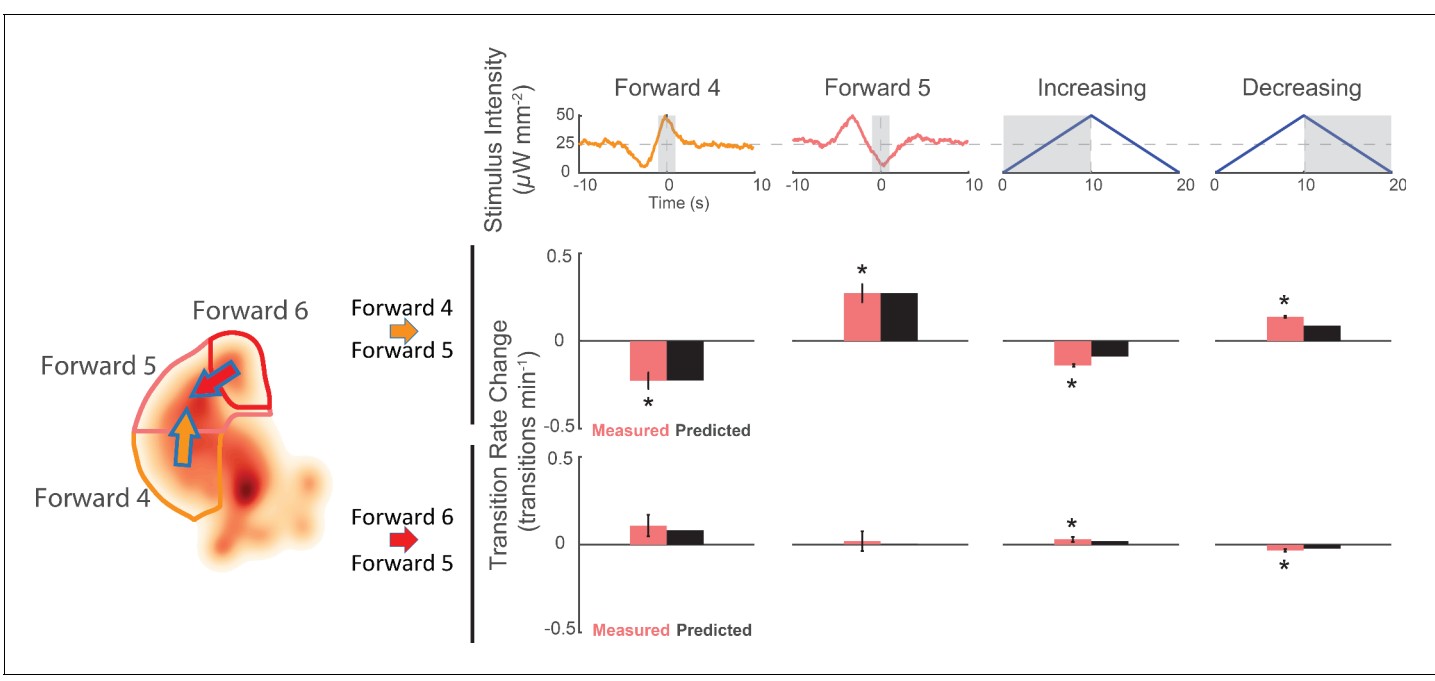

**Figure 7.** Animals respond to the same stimuli differently depending on their current behavior state. The change in transition rate from baseline is shown for transitions into 'Forward 5' from either 'Forward 4 '(middle row) or 'Forward 6' (bottom row) in response to four different stimuli (columns). Observed transition rates (colored bars) are compared to LN model predictions (black bars). The stimulus affects the rate of transitions into 'Forward 5' differently depending on whether the animal was in 'Forward 4' or 'Forward 6' at the time of stimulus. For example, consistent with the animal responding to a slowing-down signal, the 'Forward 4'-shaped stimulus decreases 'Forward 4'→'Forward 5' transitions, but increases 'Forward 6'→'Forward 5' transitions. A star indicates a significant change in transition rate from baseline. Gray shaded regions indicate the time windows over which the transition rate is calculated. Baseline is defined slightly differently for the kernel-shaped stimuli compared to the triangle waves (see 'Materials and methods'). Of 13,612 and 14,699 stimulus-animal presentations for 'Forward 4' and 'Forward 5' kernel-shaped stimuli, and 340,757 stimulus-animal presentations for the triangle wave, the following number of transitions were observed: 26, 24, 2,604 and 2,634 for 'Forward 4'→'Forward 5' (top row) and 6, 7, 713 and 791 for 'Forward 6'→'Forward 5' (bottom row). A t-test was used to test for signifiant changes from baseline and the following p-values were observed: 5.5e–5, 6.9e–6, 1.8e–19 and 5.1e–48 for 'Forward 4'→'Forward 5' (top row) and 6.3e–2, 8.7e–1, 3.9e–2 and 6.7e–5 for 'Forward 6'→'Forward 5' (bottom row). Error bars show the standard error of the mean.
DOI: https://doi.org/10.7554/eLife.36419.028

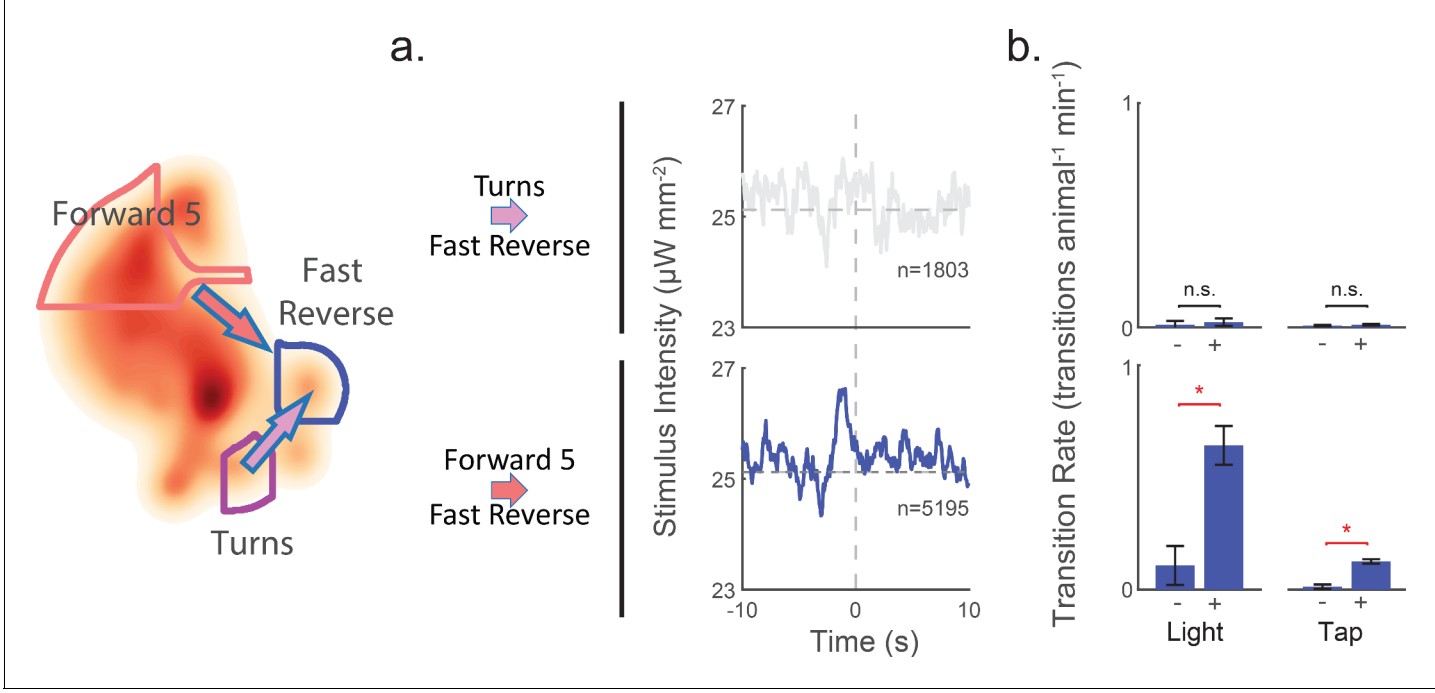

**Figure 8.** Attention to mechanosensory signals depends on behavior. When the animal is in the 'Turn' state, it ignores mechanosensory stimuli. (a) Kernels are shown for two context-dependent transitions into 'Fast Reverse'. Transitions into 'Fast Reverse' originating from 'Forward 5' are correlated with stimulus and have a significant kernel, whereas those originating from 'Turn' are not correlated with stimulus and fail our shuffled significance threshold (see methods'). The kernels shown are same as those in *Figure 6—figure supplement 1*. (b) Transition rate in response to light and tap are shown. Animals in the 'Turn' state show no significant change in transition rates in response to light or tap, whereas animals in other states, such as 'Forward 5', do show a response. The 2 s post-stimulus mean transition rate into 'Fast Reverse' is shown in response to a 1 s light stimulation (+), mechanical tap (+) or a mock control (–). A star indicates significance, calculated using an E-test (see 'Materials and methods'). Error bars show the standard error of the mean. 2,487 and 37,000 stimulus-animal presentations were analyzed for light (+) and tap (+) respectively, and 2,427 and 40,012 mock controls(–) for light and tap. The following number of transitions were observed: 1, 2, 11 and 15 for 'Turns'→'Fast Reverse' (top row) and 9, 55, 18 and 160 for 'Forward 5'→'Fast Reverse' (bottom row). P-values for the E-test are 0.68 and 0.34 for 'Turn'→'Fast Reverse' (top row) and 1.96e-9 and 0 for 'Forward 5'→'Fast Reverse'(bottom row).

DOI: https://doi.org/10.7554/eLife.36419.029

The following source data and figure supplements are available for figure 8:

**Source data 1.** P-values for transition rates in response to a light pulse for all pairwise transitions.
DOI: https://doi.org/10.7554/eLife.36419.032
**Source data 2.** P-values for transition rates in response to a tap for all pairwise transitions.
DOI: https://doi.org/10.7554/eLife.36419.033
**Figure supplement 1.** Transition rates in response to a light pulse for all pairwise transitions.
DOI: https://doi.org/10.7554/eLife.36419.030
**Figure supplement 2.** Transition rates in response to a tap for all pairwise transitions.
DOI: https://doi.org/10.7554/eLife.36419.031

instead of transitioning to a stimulus-defined velocity. This is of interest because it implies a form of context dependency: it suggests that the same stimulus will drive the animal into forward locomotory states of different speeds depending on the animal's current state.

To determine whether the stereotyped speed-up or slow-down stimulus does indeed cause the animal to speed up or slow down, we again inspected the animal's response to the kernel-shaped stimuli or the triangle-wave stimulus. Indeed, we found that the same stimulus drives the animal into a different forward locomotory state depending on the animal's current state (see *Figure 7*). For example, animals in the slower 'Forward 4' state responded to a 'Forward 4' kernel-shaped stimulus by *decreasing* their transitions to 'Forward 5'. By contrast, animals in the faster 'Forward 6' state responded to the same stimulus by *increasing* their transitions into 'Forward 5'. This was one of multiple instances in which we observed the animal responding to the same stimuli with opposite

responses depending on its current behavior. During triangle wave stimulation, for example, an increasing ramp causes slowing down, whereas a decreasing ramp causes speeding up (see *Figure 7*). We therefore conclude that stereotyped mechanosensory signals drive the animal to speed up or slow down.

## Attention to mechanosensory signals depends on the animal's current behavior

When the animal turns, it ignores all tested mechanosensory signals. This surprising observation is predicted by reverse-correlation analysis and confirmed by optogenetic and tap stimulation. Transitions out of 'Turn' are uncorrelated with stimulus, and kernels for those transitions all fail to pass our shuffled significance threshold (see bottom row in *Figure 6—figure supplement 1*). Consequently, the kernels predict that the animal should ignore mechanosensory stimuli during turns. By contrast, for every other behavior state, there is always at least one (and often many) transitions exiting out of the state whose kernels pass our significance threshold (all rows other than 'Turn' have at least one significant kernel).

To further test whether the animal does indeed ignore stimuli during turns, we investigated the animal's context-dependent response to light pulses or tap. When the animal was in the 'Turn' state, neither a light pulse nor a tap evoked a significant change in the rate of transitions into any other behavior (see bottom row *Figure 8—figure supplements 1* and *2*) (multiple-hypothesis corrected E-test, see 'Materials and methods'). By contrast, when the animal was in other states, such as 'Forward 5', both a tap and light pulses evoked significant changes in the transition rate into other behaviors. In fact, every other behavior state except for 'Forward 2' had at least one behavior transition exiting the state whose transition rate was significantly affected by either light or a tap. The 'Turn' behavior state was unique in that none of the kernels for transitions originating in 'Turn' passed the shuffled significance threshold, and no transition rates changed significantly in response to either light or a tap, (see *Figure 8—figure supplements 1* and *2*). We therefore conclude that in 'Turn, but not other states, the animal ignores mechanosensory stimuli.

Transitions into 'Fast Reverse' provide an illustrative example (see *Figure 8*). When the animal is in the 'Turn' state, there is no significant difference in the rate of 'Turn'→'Fast Reverse' transition between shuffled and stimuli. But when the animal is in 'Forward 5', light and taps caused a significant increase in 'Forward 5'→'Fast Reverse'. Taken together, we conclude that the animal attends to mechanosensory signals during most behavior states, such as 'Forward 5', but ignores them during turns.

## Discussion

This work provides new insights into *C. elegans* sensory processing. First, we show that the animal's behavioral response is tuned to the temporal properties of mechanosensory signals, such as the derivative, that extend over many seconds in time. Moreover, by adjusting the waveform of a stimulus, mechanosensory signals in the same neurons can be tailored to elicit different behavioral responses. Second, mechanosensory signals influence a broad set of behaviors. Mechanosensation not only drives reversals and accelerations but can also evoke the animal to slow down. Third, even short timescale behavior states can influence the animal's sensory processing. Earlier work has reported context-dependent sensory processing for behaviors with timescales of minutes to hours, such as hunger-satiety (*Ghosh et al., 2016*), quiescence (*Raizen et al., 2008*; *Schwarz et al., 2011*; *Nagy et al., 2014b*; *Cho et al., 2018*), arousal (*Cho and Sternberg, 2014*) or Dauer formation (*Chen and Chalfie, 2014, 2015*). Here, we show that seconds-long timescale behaviors can also profoundly alter how the animal responds to a stimulus. Most dramatically, when the animal turns, it appears to ignore mechanosensory signals completely.

A high throughput approach was crucial in revealing these new findings. Previously, we had probed the behavior response to mechanosensation using a targeted illumination system that allowed us to probe individual mechanosensory neuron pairs (*Leifer et al., 2011*). That approach, however, is impractical for collecting the thousands of animal-hours of recordings needed here. In this work, we instead activate all mechanosensory neurons simultaneously, which allows us to study many animals in parallel. The set of neurons that is activated is determined by the opsin's expression pattern. If opsins were expressed only in a single neuron, the current approach would also achieve

single-neuron resolution. Although single-cell promoters are not known for any of the soft-touch mechanosensory neurons, intersectional approaches may allow the targeting of subsets (*Wei et al., 2012*; *Schmitt et al., 2012*). Future work is needed to explore the role of individual mechanosensory neurons in temporal processing.

Automated behavior mapping was also critical for interpretation of the thousands of hours of animal behavior. We chose to classify behavior into discrete states, which are a natural description of discrete behaviors such as turns or reversal events. Alternatively, one could have chosen to use continuous description of behaviors, such as velocity, angular velocity or acceleration, which might be a more natural description of forward locomotion and speeding up or slowing down.

The linear-nonlinear (LN) model was used to map out the relationship between sensory signals and behavior, and it predicts the animal's response to stimuli satisfactorily. The LN model was chosen largely because of its ubiquity in neuroscience and simplicity of interpretation. We suspect that other models would yield similar findings. The LN model assumes a particular structure of linear and non-linear processing that is not inherently motivated by the biology, and it fails to take into account longer-timescale effects such as habituation. By contrast, the Gated Recurrent Unit (GRU) neural network model is one example of a model that is entirely non-linear and known to handle multi-timescale dependencies (*Cho et al., 2014*). GRUs are just one of many alternative models with varying degrees of complexity and interpretability that could be used to probe temporal processing (*Glaser et al., 2017*).

In more complex sensory systems such as the retina, we have come to expect that the nervous system is carefully tuned to the temporal properties of sensory signals (*Meister and Berry, 1999*). Recently, it was shown that in drosophila, temporal processing is important for behavioral responses to odor, light and sound (*Behnia et al., 2014*; *Coen et al., 2014*; *Gepner et al., 2015*; *Hernandez-Nunez et al., 2015*). And in the much simpler *C. elegans*, temporal processing within timescales in the order of seconds has been observed in thermosensation (*Clark et al., 2006, 2007*), as well as in chemosensation (*Kato et al., 2014*) where it is known to be crucial for guiding thermotaxis or chemotaxis. In the *C. elegans* mechanosensory circuit, it had been shown previously that temporal processing occurs at the receptor level in order to convert applied forces into evoked currents, with a timescale of tens of milliseconds (*Eastwood et al., 2015*), but it had remained unclear whether the nervous system used temporal information downstream to determine the animal's behavioral response. In this work, we now see evidence of temporal processing on seconds-long behavior-relevant timescales that guides the animal's behavioral response. This temporal processing may arise from recurrent activity in the neural network downstream of the touch receptor neurons. The observation of such behavior-relevant temporal processing even in the simple mechanosensory circuit raises the possibility that temporal processing may be ubiquitous across sensory systems for driving behavior.

Why might it be beneficial for the *C. elegans* nervous system to have evolved to tune its behavioral response to the temporal properties of mechanosensory signals, such as the derivative, over seconds? The natural ecology of *C. elegans* is not well understood (*Félix and Braendle, 2010*) and the statistics of the forces that it encounters in its natural environment are not known. We speculate that it could be useful for the worm to react differently if mechanosensory signals are increasing or decreasing, instead of making decisions solely on the overall stimulus strength. Note that we have characterized temporal processing to optogenetic signals, thus bypassing the animal's mechanoelectro transduction machinery. Further work is needed to characterize the temporal processing of applied forces directly.

It is is striking and surprising that the animal ignores mechanosensory inputs during turning. Why might the animal have evolved to ignore such signals during turns? The turn is part of the *C. elegans* escape response, an avoidance behavior that shares some similarities with escape responses in other organisms, such as crayfish, mollusks or goldfish (*Pirri and Alkema, 2012*). *Caenorhabditis elegans* escape consists of reverse locomotion, followed by a turn and then forward locomotion in a new direction. The turn allows the animal to reorient and navigate away from a predator, and defects in this circuit have been shown to decrease survivability (*Maguire et al., 2011*). Failing to complete the turn could inadvertently cause the animal to retrace its steps and return to danger.

Ultimately, we see evidence of two kinds of internal processes that govern how the animal interprets sensory signals. First, the animal integrates mechanosensory information over a timescale of seconds. Second, the animal interprets these signals differently depending on the animal's behavior

state. An exciting future direction will be to identify the neural circuit mechanisms that allow the worm's nervous system to integrate mechanosensory signals over time; and to alter its response rapidly depending on behavior state. This could shed insight into how internal brain states rapidly modulate sensory processing in a simple model system.

## Materials and methods

### Strains

The two strains used in this study were wild-type N2 Bristol animals (RRID:WB-STRAIN:N2_(ancestral)) and AML67 (RRID:WB-STRAIN:AML67) (wtfls46[p*mec-4::Chrimson::SL2::mCherry::unc-54*]), a transgenic strain that expresses the light-gated ion channel Chrimson and a fluorescent protein mCherry in mechanosensory neurons. To generate AML67, 40 ng of plasmid (pAL::p*mec-4::Chrimson::SL2::mCherry::unc-54*) were injected into N2 animals and integrated via UV irradiation (*Evans, 2006*). These animals were outcrossed with N2 six times. AML67 has been deposited in the public Caenorhabditis Gentics Center repository at the University of Minnesota. Plasmid pAL::p*mec-4::Chrimson::SL2::mCherry::unc-54* (https://www.addgene.org/107745/) was engineered using a HiFi Cloning Kit (NEB). Chrimson was a kind gift from Ed Boyden of MIT. mCherry and backbone was amplified from pJIM20, a gift from John Murray of the University of Pennsylvania. The promoter sequence (mec-4), splicing sequence (SL2) and 3′-utr sequence (*unc-54*) were amplified using primers as listed in *Table 1*. The construct was sequenced confirmed before injection.

Transgenic animals exhibited reduced sensitivity to a tap or touch compared to wild-type animals, presumably because Chrimson competes with endogenous MEC-4 protein for transcription (see *Figure 2—figure supplement 4*). From the alleles we had generated, we selected AML67 for use in this study because it was the most sensitive to tap and touch, despite being reduced compared to wild-type.

### Nematode handling

Strains were maintained on 9 cm NGM agar plates seeded with OP50 *Escherichia coli* food at 20° C . Worms were bleached 3 days prior to the experiment to provide 1-day-old adults. For optogenetic experiments, bleached worms were placed on plates seeded with 1 ml of 0.5 mM all-trans-retinal (ATR) mixed with OP50. Control plate lacked ATR. To avoid inadvertent optogenetic activation, plates were wrapped in aluminum foil, handled in the dark, and viewed under dissection microscopes using dim blue light.

To harvest worms for high-throughput experiments, roughly 100 to 200 worms were cut from agar, washed and then spun-down in a 1.5 ml micro centrifuge tube. Worms at the bottom of the tube were placed on an unseeded 9 cm NGM agar plate via a micropipette. Excess liquid on the

**Table 1.** Forward and reverse primer sequences used to generate pAL::p*mec-4::Chrimson::SL2::mCherry::unc-54*.

| Primer | Sequence |
| --- | --- |
| mec-4_fwd | AAGCTTCAATACAAGCTC |
| mec-4_rev | TAACTTGATAGCGATAAAAAAAATAG |
| CHRIMSON_fwd | ATGGCTGAGCTTATTTCATC |
| CHRIMSON_rev | AACAGTATCTTCATCTTCC |
| SL2_fwd | GGTACCGCTGTCTCATCC |
| SL2_rev | GATGCGTTGAAGCAGTTTC |
| mCherry_fwd | ATGGTCTCAAAGGGTGAAG |
| mCherry_rev | TTATACAATTCATCCATGCC |
| U54_fwd | GCGCCGGTCGCTACCATTAC |
| U54_rev | AAGGGCCCGTACGGCCGA |

DOI: https://doi.org/10.7554/eLife.36419.034

plate was carefully wicked away using tissue paper. Worms were allowed to adapt to their new environment for 25 min before recordings or stimulation were carried out.

## High-throughput imaging

Experiments were conducted in a custom-built high-throughput imaging rig (*Figure 2—figure supplement 1*). Plates of animals were recorded while undergoing 30 min of optogenetic or tap stimulation. Imaging was performed as follows: the agar plate was illuminated by a ring of 850 nm infrared LEDs (irrf850-5050-60-reel, environmentallights.com). A 2592 × 1944 pixel CMOS camera (ACA2500-14um, Basler) recorded worm movements at 14 frames per second and a magnification of 20 µm per pixel, so as to provide sufficient spatiotemporal resolution to capture posture dynamics. The field of view of the camera was centered on the plate and included approximately 50% of the plate surface. Custom LabVIEW software acquired images from the camera and controlled stimulus delivery as described below.

## Tap delivery

Taps were delivered to the side of 9 cm plates containing the animals by means of a solenoid, following a method similar to that described by *Swierczek et al. (2011)*. An electric solenoid tapper (Small Push-Pull Solenoid, Adafruit) was driven with a 70 ms, 24 V, DC pulse under Labview control via a LabJack DAQ and a solid-state relay. During tap experiments, taps were delivered to the plate once per minute for 30 min (see *Table 2*). The 1 min inter-stimulus interval was chosen to minimize habituation (*Timbers et al., 2013*).

## Optogenetic stimulation

Experiments involving optogenetic stimulation are summarized in *Table 2*. Optogenetic stimulation was delivered by three 625 nm LEDs (M625L3, Thorlabs) positioned such that their light approximately tiles the agar plate visible in the camera's field of view. LED's were driven by a diode driver (L2C210C, Thorlabs) under the control of LabVIEW via an analog signal from a LabJack DAQ (Model U3-HV with LJTick-DAC). The range of the light intensity for optogenetic stimulation averaged at the plate spanned from 0 to 80 µW mm$^{-2}$. Small spatial inhomogeneities in light intensity were characterized and accounted for in software so as to calculate the precise light intensity delivered to each animal. An infared long pass filter (FEL0800, Thorlabs) in front of the camera blocked light from the stimulus LEDs and only permitted light from the infrared behavior LEDs.

### Optogenetic pulse stimulus

For optogenetic pulse experiments, as in *Figure 2*, a 1 s light pulse was delivered once per minute for 30 min. Initial experiments measured the behavioral responses to pulses of different light intensities. In those experiments, shown in *Figure 2c*, the light intensity of the pulse was randomly shuffled such that five pulses each of 2, 5, 10, 50, and 80 µW mm$^{-2}$ were delivered during the 30 min recording.

### Random noise optogenetic stimulus

Experiments involving reverse correlation all used a light stimulus with intensity modulated by random broad-spectrum noise. The random noise stimulus was generated according to,

$$s(t+1) = As(t) + Bn_{\mathrm{rand}} + C, \tag{2}$$

where $A \equiv exp - \left( \tau_{\mathrm{period}} / \tau_c \right)$ and $B \equiv \sigma_{\mathrm{rms}} \sqrt{1 - A^2}$. Here $s(t+1)$ is the stimulus intensity at the next time-point, $A$ is the weighting of the previous stimulus $s(t)$, $B$ is the weighting of a random number, $n_{\mathrm{rand}}$, drawn from a Gaussian distribution with standard deviation given by $\sigma_{\mathrm{rms}}$, and $C$ is a constant offset that sets the average stimulus intensity. The weighting $A$ is related to correlation time $\tau_c$ and the duration of our time step $\tau_{\mathrm{period}}$. Because in our setup the stimulus is updated with each image acquisition, the time step $\tau_{\mathrm{period}}$ is the inverse of the image acquisition rate, or approximately 0.07 s for 14 Hz.

Both $C$ and $\sigma_{\mathrm{rms}}$ were chosen to be 25 µW mm$^{-2}$ so that the function generated intensities that mostly fell in the intensity range of 0–50 µW mm$^{-2}$, a regime that appeared to be most sensitive to

**Table 2.** Summary of experimental conditions.

Each experimental series consisted of recordings of multiple plates usually spread across multiple days, as indicated. Recordings were all 30 mins in duration for each plate. Note that two methods were used to tally the number of stimulus-animal presentations (see 'Materials and methods'). Here, a stimulus presentation is counted even if the track was interrupted mid-presentation.

| Experiment series | Strain | Stim | ATR | Number of plates | Number of days | Interstimulus interval (s) | Stimulus duration (s) | Total animal-stimulus presentations | Cumulative Recording Length (animal-hours) | Animals per frame (Mean ± Stdev) | Figures |
|---|---|---|---|---|---|---|---|---|---|---|---|
| Random noise | AML67 | Light | + | 58 | 3 | n/a | n/a | n/a | 1,784 | 62 ± 34 | Figure 1, Figure 1—figure supplement 2, Figure 1—video 1, Figure 1—video 2, Figure 1—video 3, Figure 3, Figure 3—figure supplement 1, Figure 3—figure supplement 2, Figure 3—figure supplement 4, Figure 3—figure supplement 5, Figure 6, Figure 6—figure supplement 1, Figure 8 |
| | | | − | 20 | 3 | n/a | n/a | n/a | 500 | 50 ± 23 | Figure 1, Figure 1—figure supplement 2, Figure 1—video 2, Figure 1—figure supplement 1, Figure 3—figure supplement 1, Figure 3—figure supplement 3, Figure 3—figure supplement 5, Figure 6—figure supplement 2 |
| Triangle wave | AML67 | Light | + | 62 | 3 | 0 | 20 | 340,757 | 1,912 | 62 ± 42 | Figure 5, Figure 7 |
| | | | − | 20 | 3 | 0 | 20 | 142,461 | 800 | 80 ± 55 | Figure 5—figure supplement 1 |
| Kernel-shaped Stimuli | AML67 | Light | + | 44 | 3 | 40 | 20 | 84,875 | 1,453 | 66 ± 40 | Figure 4, Figure 4—figure supplement 1, Figure 7 |
| | | | − | 12 | 3 | 40 | 20 | 22,866 | 392 | 65 ± 33 | Figure 4—figure supplement 1 |
| Light pulse | AML67 | Light | + | 12 | 1 | 59 | 1 | 15,128 | 260 | 43 ± 24 | Figure 2, Figure 8 |
| | | | − | 6 | 1 | 59 | 1 | 8107 | 139 | 46 ± 18 | Figure 2—figure supplement 3 |
| Plate tap | AML67 | Tap | + | 7 | 2 | 60 | Impulse | 21,117 | 366 | 105 ± 60 | Figure 2—figure supplement 4 |
| | | | − | 8 | 2 | 60 | Impulse | 14,646 | 254 | 64 ± 46 | |
| | N2 | Tap | − | 22 | 3 | 60 | Impulse | 40,409 | 695 | 63 ± 25 | Figure 2, Figure 8, Figure 8—figure supplement 2 |
| Total | | | | 271 | | | | | 8,554 | 63 ± 40 | |

DOI: https://doi.org/10.7554/eLife.36419.035

behavior response (see *Figure 2c*). $\tau_c$ was chosen to be 0.5 s as this roughly matched our intuition about the timescale of temporally varying mechanical stimuli that the animal might encounter while navigating its natural environment. Finally, the stimulus was clipped and forced to stay in the range of 0–50 µW mm$^{-2}$. Frequency spectra of our stimuli are shown in *Figure 3—figure supplement 4*.

### Triangle wave optogenetic stimulus

Triangle wave stimuli were also generated. Triangle waves were linearly increasing ramps of light intensity from 0 µW mm$^{-2}$ to 50 µW mm$^{-2}$ for 10 s followed by linearly decreasing ramps of 50 µW mm$^{-2}$ to 0 µW mm$^{-2}$ for 10 s, repeated continuously for 30 min.

### Kernel-shaped (tailored) stimulus

In the tailored stimulation experiments, stimuli were generated from the behavior-triggered averages found using reverse correlation. The six behavior-triggered averages from *Figure 3* were scaled in intensity until either their minimum was at 0 µW mm$^{-2}$ or the maximum was at 50 µW mm$^{-2}$. These were then shuffled and played back one per minute such that each behavior-triggered average was delivered 5 times per 30 min experiment. 25 µW mm$^{-2}$ of constant light intensity was delivered between stimulus presentation.

## Measuring animal behavior

The unsupervised behavior mapping approach used in this work is adapted from work in drosophila (*Berman et al., 2014*) and is similar in spirit to work in rodents (*Wiltschko et al., 2015*). It also builds upon decades of methodological advances quantifying *C. elegans* behavior (*Croll, 1975*; *Stephens et al., 2008*; *Ramot et al., 2008*; *Brown et al., 2013*; *Yemini et al., 2013*; *Gyenes and Brown, 2016*; *Gomez-Marin et al., 2016*).

Animal behavior was measured and classified using an analysis pipeline, summarized in *Figure 1—figure supplement 1*. First, the worms were located and tracked, then their posture was extracted, and finally their posture dynamics were clustered and classified. Details of the pipeline are described below. The pipeline was written in MATLAB and run on the Princeton University's high-performance parallel computing cluster. Source code is available at (https://github.com/leiferlab/liu-temporal-processing) (Liu and Leifer, 2018; copy archived at https://github.com/elifesciences-publications/liu-temporal-processing).

### Animal location tracking

To first identify animals and to track their location, raw video of animals on plates was analyzed using a modified version of the Parallel Worm Tracker (*Ramot et al., 2008*). Animals were found via binary thresholding and centroid tracking (*Figure 1—video 1*).

### Animal posture extraction

The animal's posture was found by extracting the animal's centerline from the video using custom MATLAB scripts. Videos of each individual worm were first generated by cropping a 70 × 70 pixel region around the worm's centroid in every frame. A centerline with 20 points was fitted to the image at each frame using an active contour model similar to that used by *Nguyen et al., 2017*), which was inspired by the one described by *Deng et al. (2013)*. The algorithm for fitting the centerline was specifically optimized to measure the posture of the worm in a variety of conditions, including when the animal crossed over itself during turns. The active contour model fits the centerline by relaxing contiguous points along a gradient defined by four forces: (1) an image force that fits the contour to the image of the worm; (2) a tip force that guides the beginning and end of the contour to the worm's presumptive head and tail; (3) a spring force that guides the contour to be similar lengths; (4) and a repel force that makes sure that the contour does not stick to itself. To ensure continuity in time, the active contour of the following frame is initialized by the relaxed contour of the previous frame. The head and the tail of the worm were determined by assuming that the worm moves forward the majority of the time. A quality score was calculated to estimate how well the centerline fit the image and how much it displaced from the previous centerline. On the rare occasion when the quality score of a frame fell below threshold, that frame was dropped, and the track was split into two.

## Posture dimensionality reduction

To interpret the animal's posture more efficiently, the dimensionality of the animal's centerline was reduced from 20 position $(x, y)$ coordinates to five posture coefficients using principle component analysis (PCA), following the method of *Stephens et al. (2008)*. Principle components of posture were extracted from recordings of approximately 2 million animal-frames of freely behaving N2 worms. Centerlines were converted into a series of angles oriented such that the mean angle is 0. The first five principle components explain >98% of the posture variance. The animal's posture dynamics were thus represented as a time-series of five coefficients, one for each of the five principle posture modes.

## Generating spectrograms of posture dynamics

To characterize posture dynamics, a spectrogram was generated for each of the posture mode coefficients (as in *Berman et al. [2014]*). A Morlet continuous wavelet transform was performed on each of the five coefficient time series at 25 frequencies dyadically spaced between 0.3 Hz and 7 Hz. The low-frequency bound was chosen to reflect our intuition regarding the timescale of *C. elegans* behavior and the high-frequency bound was set by the Nyquist sampling frequency of our image acquisition. The spectrogram provides information about the frequency spectra of the animal's posture dynamics but it lacks information about the phase of the animal's posture, which is important for discerning forward from backward locomotion. To preserve forward and backward information, we introduced a binary 'directionality' vector that is 2 when the worm centroid is moving forward, and 1 when the worm centroid is backwards. Directionality was calculated by taking the sign of the dot product of the head vector with a tangent vector of the animal's centroid trajectory. Together, the five spectrograms and directionality vector provide a 126 dimensional feature vector that describe the animal's behavior at each time point. It is this feature vector that is clustered, as described below.

## Defining the behavioral map and behavior states

To classify behavior into discrete stereotyped behavior states that emerge naturally from our recordings, we followed a behavior-mapping strategy described in *Berman et al. (2014)*. A single behavior map was generated so that behaviors were defined consistently across all experiments. To generate the behavior map, 50,000 animal-time points were uniformly sampled from the 2,284 animal-hours of behavior recordings made during random-noise optogenetic stimulation. Each animal-time point contributes a 126-dimensional feature vector describing the animal's instantaneous behavior. We generated a two-dimensional map of these feature vectors by embedding the 126-dimensional space in a plane using a non-linear dimensionality reduction technique called t-distributed stochastic neighbor embedding (t-SNE) (*Lvd and Hinton, 2008*). Under t-SNE, each feature vector is embedded such that the local distance between feature vectors is conserved but long distance scales are distorted (*Figure 2—figure supplement 2a*).

We then generated a probability density histogram of behavior by projecting all $10^8$ behavior time points from the 2,284 animal-hours of random noise optogenetic stimulation (*Figure 1—figure supplement 2b*) into the 2D map. Clusters of high probability in this density map corresponded to a distinct stereotyped behavior. Stereotype behaviors were defined by water-shedding the probability density map (*Figure 1—figure supplement 2c*) and each region was assigned a name such as 'Forward 3' (*Figure 1b*). Videos showing examples of worms exhibiting behaviors in each region are shown in *Figure 1—video 2*. Time points from subsequent recordings were similarly projected into this map for the purposes of classifying animal behavior.

## Identifying behavioral transitions

At each time point, the worm belongs to a point in the 2D behavior map described above (see *Figure 1—video 3*). Animals that dwelled in one behavior region for at least 0.5 s were classified as exhibiting that behavior during all contiguous time points in that behavior region. Animals inhabiting a behavior region for less than 0.5 s were classified as 'in transition'.

A transition into behavior $X$ is defined to occur on the first time point that the animal is classified as in $X$. Transitions from behavior $W \rightarrow X$ were defined to occur on the first time point the animal is classified as in $X$ provided that: (i) the animal transitioned directly from $W$ to $X$; or (ii) the animal had

previously been classified as in $W$, was then classified as 'in transition', and then was classified as in state $X$. Cases where the animal was in $X$, then 'in transition' and then returned to $X$, were ignored.

## Ambiguities in temporal definition of behavior

The wavelet spectrogram introduces an inherent uncertainty in the precise timing of a behavior transition. This ultimately arises from the uncertainty principle: behavior dynamics that have low-frequency components provide less temporal resolution than higher-frequency dynamics. An equivalent view is that the spectrogram feature vector at any given moment is influenced by temporally adjacent postural dynamics in the past and future, and this influence is stronger at lower frequencies than at higher ones.

This temporal uncertainty or 'bleeding over' of future behavior, causes the animal occasionally to appear to respond (but not actually respond) to a stimulus prior to its delivery. In the worst case, the time-scale of this leakage is set by our choice of the lowest frequency wavelet, which is 0.3 Hz (i.e. 2.7 s). Behaviors with strong higher-frequency components have shorter timescale uncertainties. We take large time windows of 20 s to define our kernels; in part, so that a few second time-shift does not result in any loss of information.

## Reverse correlation

Reverse correlation was used to find a linear kernel and non-linearity that describe the relationship between the animal's behavior transitions and an applied stimulus.

### Calculating kernels

Linear kernels for each behavior were estimated by computing the behavior-triggered average of the stimulus,

$$\hat{A} = \frac{1}{N} \sum_{n=1}^{N} \vec{s}(t_n),$$

(3)

where $t_n$ is the time of $n$th behavioral transition, $\vec{s}(t_n)$ is a vector representing the stimuli presented during a 20 s temporal window around $t_n$, and $N$ is the total number of behavioral transitions (*Schwartz et al., 2006*). The linear kernel was estimated to be the mean-subtracted, time-reversed behavior-triggered average.

### Kernel significance

Behavior-triggered averages (also referred to as kernels) were deemed significant if their magnitude (L2 norm) exceeded the top 1 percent of a distribution of random kernels found by shuffling the stimulus in time. Shuffling was performed in such a way as to preserve the temporal properties of the transition train while completely decorrelating it from the stimulus. Specifically, shuffling was performed by circle-shifting the transition timings within every track by a randomly selected integer between one and the number of time points in the track. Shuffled kernel distributions for each behavior were generated by recalculating the behavior-triggered average 100 times, each with different circle-shifted timings.

### Estimating the non-linearity

The non-linearity $f$ allows the probability of a behavior transition to be estimated from the filtered signal, namely the stimulus convolved with the linear kernel (*Gepner et al., 2015*). Non-linearities were estimated from the ratio of two histograms: the first is a histogram of time-point counts versus filtered signal given a behavioral transition at that time-point, and the second is a histogram of time-point counts versus filtered signal for all time-points (*Schwartz et al., 2006*). Histograms were tabulated with 10 equally spaced bins spanning the range of the filtered signal. Bin-wise division of the two histograms yielded 10 points relating probability of behavior to filtered signals (*Figure 3—figure supplement 2*). For each point, we calculate a propagated error, $E$, assuming Poisson counting statistics,

$$E = \sqrt{\frac{T-1}{F^2} + \frac{T^2(F-1)}{F^4}}, \tag{4}$$

where $T$ is the number of behavioral transitions in that bin, and $F$ is the number of filtered signal time-points in that bin. We then fitted a two parameter exponential to the 10 points, weighing each point by the inverse of the error in order to reduce the influence of noise. This fitted exponential function is our estimate of the non-linearity.

## Calculating transition rates

When presented as a timeseries of rates, as in *Figure 2—figure supplement 2*, transition rates were calculated according to the following: behavior timeseries from all recordings were cropped in a time window around each stimulus, commingled, and then time aligned to the stimulus. The fraction of all animals undergoing a transition was calculated at each time step. The fractions of animal were directly converted into a rate of transitions per animal per minute, yielding the timeseries of rates.

### Calculating transition rate changes

Transition rate change, as in *Figure 4b*, *Figure 4—figure supplement 1*, *Figure 5*, and *Figure 7*, were calculated as follows: an average transition rate was found in a time window during a stimulus (as described above), and then a baseline was subtracted off. For kernel-shaped stimuli experiments (*Figure 4a* and *Figure 4—figure supplement 1*), the baseline is defined as the average transition rate in a 20 s time window prior to each stimulus. For the triangle wave in *Figure 5*, the baseline was defined to be the overall mean transition rate throughout the recording.

In cases where a bar is shown (*Figure 4a*, *Figure 7*), a change in transition rate was calculated by averaging the timeseries of rates over a time window (indicated in the those figures by gray shading).

### Measuring transition rates for context-dependent tap or light-induced experiments

Transition rates were calculated slightly differently in *Figure 8* and *Figure 8—figure supplements 1* and *2* to facilitate significance testing via the E-test (*Krishnamoorthy and Thomson, 2004*). The transition rate in a 2 s time window immediately following light pulse or tap (+) was compared to a transition rate in a 2 s window immediately following a mock control (–). Mock controls were set to occur at the mid point between consecutive stimuli.

Instead of calculating the transition rates at each time bin and then averaging across time, as described previously, we instead calculated a single transition rate for the entire 2 s time window by comingling transitions from all time bins, as follows. We 1) selected tracks that were uninterrupted for the 2 s, (2) counted the first transition (if it occurred) within 2 s after stimulus onset across all of our experiments, (3) divided by the total number of tracked time points, and (4) converted the value to transitions per animal per minute. The number of stimulus-animal presentations differs slightly from those in *Figure 2* because now tracks are required to be contiguous for 2 s after stimulus presentation, which was not a requirement previously.

P-values were attained using an E-test (*Krishnamoorthy and Thomson, 2004*). To account for testing 72 behavior transitions concurrently, we use the Bonferroni multiple-hypothesis correction. Only p-values less than $\alpha = 0.05/72 = 7 \cdot 10^{-4}$ are considered significant.

In our analysis of light-pulse response, we grouped all stimulation light intensities together.

## Data

Behavioral analysis and stimulation data for all tracked animals in all experiments in *Table 2* are available at https://doi.org/10.6084/m9.figshare.5956348. See dataset README for details. All recorded data, including raw images (2 TB), will be available at http://dx.doi.org/10.21227/H27944.

## Acknowledgements

We thank Marc Gershow (NYU) and Gordon Berman (Emory) for productive discussions and trouble-shooting. We also thank Mala Murthy and Jonathan Pillow, both of Princeton University, for

productive discussions and feedback. Alicia Castillo Bahena and Tayla Duarte contributed to preliminary studies of this work. This work was supported by grants from the Simons Foundation (SCGB #324285 and SCGB #543003, to AML). This work was also supported by Princeton University's Dean for Research Innovation Fund (to AML and JWS). Research reported in this publication was supported by the National Human Genome Research Institute of the National Institutes of Health under Award Number T32HG003284. Some strains were provided by the CGC, which is funded by NIH Office of Research Infrastructure Programs (P40 OD010440). The content is solely the responsibility of the authors and does not necessarily represent the official views of the National Institutes of Health.

# Additional information

## Funding

| Funder | Grant reference number | Author |
| --- | --- | --- |
| National Institutes of Health | National Human Genome Research Institute Award Number T32HG003284 | Mochi Liu |
| Princeton University | Dean for Research Innovation Fund | Joshua W Shaevitz Andrew M Leifer |
| National Science Foundation | 1734030 | Joshua W Shaevitz Andrew M Leifer |
| Simons Foundation | SCGB #324285 | Andrew M Leifer |
| Simons Foundation | SCGB #543003 | Andrew M Leifer |

The funders had no role in study design, data collection and interpretation, or the decision to submit the work for publication.

## Author contributions

Mochi Liu, Conceptualization, Software, Formal analysis, Investigation, Methodology, Writing—original draft, Writing—review and editing; Anuj K Sharma, Resources, Methodology, Writing—review and editing, Performed all transgenics; Joshua W Shaevitz, Conceptualization, Supervision, Funding acquisition, Writing—review and editing; Andrew M Leifer, Conceptualization, Supervision, Funding acquisition, Writing—original draft, Project administration, Writing—review and editing

## Author ORCIDs

Anuj K Sharma  http://orcid.org/0000-0001-5061-9731
Andrew M Leifer  http://orcid.org/0000-0002-5362-5093

## Decision letter and Author response

Decision letter https://doi.org/10.7554/eLife.36419.042
Author response https://doi.org/10.7554/eLife.36419.043

# Additional files

## Supplementary files

• Transparent reporting form
DOI: https://doi.org/10.7554/eLife.36419.036

## Data availability

Stimulus and behavior data have been made publicly available on Figshare https://doi.org/10.6084/m9.figshare.5956348. Raw imaging data (2TB) have been made publicly available on IEEE DataPorts http://dx.doi.org/10.21227/H27944.

The following datasets were generated:

| Author(s) | Year | Dataset title | Dataset URL | Database, license, and accessibility information |
|---|---|---|---|---|
| Liu M, Sharma AK, Shaevitz JW, Leifer AM | 2018 | Temporal processing and context dependency in C. elegans mechanosensation dataset | https://doi.org/10.21227/H27944 | Publicly available on IEEE DataPorts (DOI: 10.21227/H27944) |
| Liu M, Sharma AK, Shaevitz JW, Leifer AM | 2018 | Temporal processing and context dependency in C. elegans mechanosensation stimulus and behavior dataset | https://doi.org/10.6084/m9.figshare.5956348 | Available at figshare under a CC0 Public Domain licence (https://figshare.com/) |

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
