## [Decision Letter]

Thank you for submitting your article "Temporal processing and context dependency in *C. elegans* mechanosensation" for consideration by *eLife*. Your article has been reviewed by two peer reviewers, and the evaluation has been overseen by a Reviewing Editor and Eve Marder as the Senior Editor. The reviewers have opted to remain anonymous.

The reviewers have discussed the reviews with one another and the Reviewing Editor has drafted this decision to help you prepare a revised submission. We hope you will be able to submit the revised version within two months.

Summary:

Using a combination of optogenetics and high-throughput automated behavioral segmentation in freely behaving *C. elegans*, this study demonstrates that temporal properties of mechanosensory neuron stimulation is relevant for behavior. Behavioral context (e.g. turning) affects the integration of mechanosensory signals. Together the technical approach as well as the findings provide a clear and quantitative account of how mechanosensory signals influence, and are influenced by, swimming behavior in an intact animal.

Essential revisions:

The reviewers commented on how the findings are presented in the context of what is known in the field. Specifically, the presentation gives the impression that prior studies have somehow gotten most everything wrong and that the current study completely overturns previous work. The manuscript should be tempered and more even-handed about the treatment of previous work, emphasizing important connections as well as key differences. The authors should also be more self-critical about the limitations of their approach, e.g. all of the stimuli delivered (both mechanical 'tap' stimulation and optical stimuli) are expected to simultaneously activate the entire ensemble of touch receptor neurons. Claims about ethological relevance of the experiments should also be more circumspect and spell out possible confounds.

Reviewers felt that more should be done to justify the use of the LN model – other models that can account for stochasticity may be more appropriate and should be considered/rejected explicitly.

The authors should comment on what can account for the seconds-long effects of stimulation, because direct measurements of the mechanosensory neurons show that their response occurs on a millisecond timescale.

---

## [Author Response]

Summary:Using a combination of optogenetics and high-throughput automated behavioral segmentation in freely behaving C. elegans, this study demonstrates that temporal properties of mechanosensory neuron stimulation is relevant for behavior. Behavioral context (e.g. turning) affects the integration of mechanosensory signals. Together the technical approach as well as the findings provide a clear and quantitative account of how mechanosensory signals influence, and are influenced by, swimming behavior in an intact animal.Agreed. Thank you.Essential revisions:The reviewers commented on how the findings are presented in the context of what is known in the field. Specifically, the presentation gives the impression that prior studies have somehow gotten most everything wrong and that the current study completely overturns previous work. The manuscript should be tempered and more even-handed about the treatment of previous work, emphasizing important connections as well as key differences.

We have revised the text to be more reserved. We have softened or outright removed contrasts between our work and the literature and we now highlight more positive connections. Many of these changes are in the Introduction and Discussion section, while others are sprinkled throughout the manuscript. To more easily identify textual changes, we have uploaded a separate “track-changes"-like document that shows removed text in red. We have also added in references to highlight additional work in the area. We added references exploring long time-scale modulation of touch response for example due to vibration or Dauer formation (Chen and Chalfie, 2014, Chen and Chalfie, 2015) and lethargus (Raizen et al., 2008), to the Introduction. We also now point to additional recent work probing spatial tuning to touch (McClanahan et al., 2017) and response to ultrasound (Kubanek et al., 2018), to the Introduction. We highlight work showing the slow intracellular calcium response observed in the touch neurons, (Suzuki et al., 2013; Cho et al., 2018). And we point to (Liu et al., 2009) and (Narayan et al., 2011) to justify why we expect modulating light intensity will elicit smooth time-varying membrane potentials.

The authors should also be more self-critical about the limitations of their approach, e.g. all of the stimuli delivered (both mechanical 'tap' stimulation and optical stimuli) are expected to simultaneously activate the entire ensemble of touch receptor neurons.

We have added text to the Discussion section that highlights some of the design decisions we made in our experiment and the inherent trade-offs that arise from those decisions. For example, we further emphasize that all neurons are activated simultaneously, and that future work is needed to probe the role of individual neurons in temporal processing. We also now remind the reader once more that we have characterized temporal processing in response to optogenetic stimulation, thus bypassing the animal's natural mechanoelectric transduction machinery. “Further work,” we write, “is necessary to probe temporal processing of applied forces directly.” And we highlight our choice to classify behavior as discrete states and we write that we suspect continuous descriptions of behavior would likely also reveal similar results.

Claims about ethological relevance of the experiments should also be more circumspect and spell out possible confounds.

We have removed explicit claims of ethological relevance. In the Discussion section we explain that the natural ecology of *C. elegans* is not well understood (Felix and Braendle, 2010) and the statistics of force stimuli in its natural environment are not known. We have, however, left intact the paragraph, in the Discussion section, describing turning and prey avoidance because that is well documented in the literature.

Reviewers felt that more should be done to justify the use of the LN model – other models that can account for stochasticity may be more appropriate and should be considered/rejected explicitly.

We have added a paragraph to the Discussion section to clarify our motivation in choosing the LN model. We now explain that the LN model was chosen primarily for its “simplicity and ubiquity in neuroscience” and that “we suspect that other models would yield similar findings.” In the same paragraph we now point to specific assumptions and limitations of the LN model and point to a radically different model, a GRU neural network, as one of many potential alternative models to explore.

The goal of this work is not to claim that the LN model is somehow the most appropriate, or indeed even that it is better than others. Rather, the LN model is one model that seems to work well enough. We now better convey this sentiment to the reader and we therefore would argue that further detailed comparisons to other models are unnecessary.

The authors should comment on what can account for the seconds-long effects of stimulation, because direct measurements of the mechanosensory neurons show that their response occurs on a millisecond timescale.

In the Discussion section, we now state that recurrent activity in the network downstream of the touch receptor neurons could plausibly give rise to temporal processing on a seconds scale.